# Closely related reovirus lab strains induce opposite expression of RIG-I/IFN-dependent versus -independent host genes, via mechanisms of slow replication versus polymorphisms in dsRNA binding σ3 respectively

Adil Mohamed[1], Prathyusha Konda[2], Heather E. Eaton[1], Shashi Gujar[2], James R. Smiley[1], Maya Shmulevitz[1]*

1 Department of Medical Microbiology and Immunology, Li Ka Shing Institute of Virology, University of Alberta, Edmonton, Alberta, Canada, 2 Department of Pathology and Department of Microbiology and Immunology, Dalhousie University, Halifax, Nova Scotia, Canada

* shmulevi@ualberta.ca

## Abstract

The Dearing isolate of Mammalian orthoreovirus (T3D) is a prominent model of virus-host relationships and a candidate oncolytic virotherapy. Closely related laboratory strains of T3D, originating from the same ancestral T3D isolate, were recently found to exhibit significantly different oncolytic properties. Specifically, the T3D$^{PL}$ strain had faster replication kinetics in a panel of cancer cells and improved tumor regression in an *in vivo* melanoma model, relative to T3D$^{TD}$. In this study, we discover that T3D$^{PL}$ and T3D$^{TD}$ also differentially activate host signalling pathways and downstream gene transcription. At equivalent infectious dose, T3D$^{TD}$ induces higher IRF3 phosphorylation and expression of type I IFNs and IFN-stimulated genes (ISGs) than T3D$^{PL}$. Using mono-reassortants with intermediate replication kinetics and pharmacological inhibitors of reovirus replication, IFN responses were found to inversely correlate with kinetics of virus replication. In other words, slow-replicating T3D strains induce more IFN signalling than fast-replicating T3D strains. Paradoxically, during co-infections by T3D$^{PL}$ and T3D$^{TD}$, there was still high IRF3 phosphorylation indicating a phenodominant effect by the slow-replicating T3D$^{TD}$. Using silencing and knock-out of RIG-I to impede IFN, we found that IFN induction does not affect the first round of reovirus replication but does prevent cell-cell spread in a paracrine fashion. Accordingly, during co-infections, T3D$^{PL}$ continues to replicate robustly despite activation of IFN by T3D$^{TD}$. Using gene expression analysis, we discovered that reovirus can also induce a subset of genes in a RIG-I and IFN-independent manner; these genes were induced more by T3D$^{PL}$ than T3D$^{TD}$. Polymorphisms in reovirus σ3 viral protein were found to control activation of RIG-I/ IFN-independent genes. Altogether, the study reveals that single amino acid polymorphisms in reovirus genomes can have large impact on host gene expression, by both changing

**Data Availability Statement:** All relevant data are within the manuscript and its Supporting Information files.

**Funding:** This work was funded by project grants to M.S. from the Canadian Institutes of Health Research (CIHR), Li Ka Shing Institute of Virology (LKSIoV), and Cancer Research Society (CRS), a salary award to M.S. from the Canada Research Chairs (CRC) and infrastructure support from Canada Foundation for Innovation (CFI). A.M. received scholarships from an Alberta Cancer Foundation Graduate Studentship (ACF), University of Alberta Faculty of Medicine and Dentistry/Alberta Health Services Graduate Recruitment Studentship, and University of Alberta Doctoral Recruitment Award. The funders had no role in study design, data collection and analysis, decision to publish, or preparation of the manuscript.

**Competing interests:** The authors have declared that no competing interests exist.

replication kinetics and by modifying viral protein activity, such that two closely related T3D strains can induce opposite cytokine landscapes.

## Author summary

Mammalian orthoreovirus serotype 3 Dearing (T3D reovirus) is being explored as a cancer therapy. Laboratories worldwide use independent strains of T3D, that we previously demonstrated to have different oncolytic potency *in vivo*; but what impact does the small genomic diversity between T3D reovirus strains have on host responses? Herein, we discover that host response varies significantly between T3D lab strains; for example, the T3D[TD] strain causes higher IFN induction and expression of IFN-dependent cytokines, while the T3D[PL] strain conversely causes higher induction of RIG-I/IFN-independent genes and cytokines. Mechanistically, higher IFN signalling is caused by gene polymorphisms in S4, M1, or L3 reovirus genes that decelerate virus replication kinetics, while higher non-IFN gene induction is attributed to polymorphisms in the reovirus S4 gene-encoded dsRNA binding protein σ3. Unexpectedly, IFN induction by the slow T3D[TD] strain phenodominated during T3D[TD]/ T3D[PL] coinfections, suggesting distinct behaviours of incoming virions. Moreover, IFN responses had minimal effects on the first round of reovirus replication. Overall, the data reveals how single amino acid changes can profoundly and even inversely affect host cell response to reovirus. The prospect of manipulating reovirus replication kinetics and σ3 could help optimize the host response for greatest oncolytic and immunotherapeutic potency.

## Introduction

The last two decades have brought a deep molecular understanding of how cells detect and respond to pathogen associated molecular patterns (PAMPs) resulting in production of interferons (IFNs) and other cytokines that not only induce antiviral cellular responses but also recruits and activates innate and adaptive immune cells [1, 2]. In response, viruses have evolved assorted antagonists to dampen host antiviral responses, and the role of these antagonists in dictating host range, virulence and pathogenesis is increasingly appreciated [3–6]. As iconic examples, switching of HIV tetherin antagonists is proposed to have supported cross-species adaptation [7], and myxoma virus seems to be adapting its PKR antagonist during adaptation in European rabbits [8]. What is less well understood, is the extent to which small genomic variations in a virus can impact how the host cell responds. For example, it was recently appreciated that defective viral genomes that arise during virus replication, though non-infectious, heighten host antiviral signalling [9–11]. But among infectious progeny virions, given that RNA viruses have mutation rates of approximately 1 per genome; how do single polymorphisms affect host responses?

Understanding the consequence of virus genomic diversity on host signalling will provide insights into zoonosis and pathogenesis but may also help create better therapeutic viruses. Numerous viruses are in clinical testing as cancer therapeutics. The therapeutic activity of oncolytic viruses depends on both direct cytolysis of cancer cells by the virus (i.e. evident from oncolytic activity in severely immunocompromised mice), and by stimulation of anti-tumor immunity [12]. Many studies have characterized immunological responses to oncolytic viruses and support immunotherapeutic value of tumor-specific viruses. To enhance the

immunotherapeutic response, oncolytic viruses are being explored in combination with other immunomodulatory approaches such as cytokines and immune checkpoint inhibitors [13, 14]. Given that oncolytic viruses were not adapted to trigger higher immune responses to begin with, is it surprising that their immune-activation activities are sub-par? A better understanding of how genomic variation in oncolytic viruses contributes to host response and cytokine production could lead to generation of more-immunotherapeutic oncolytic viruses.

One of the oncolytic viruses currently in phase III testing is mammalian orthoreovirus (reovirus), specifically the serotype 3 Dearing isolate (T3D). Reovirus belongs to the Reoviridae family, has 10 dsRNA genome segments, and a two-layered icosahedral capsid. Naturally, reovirus infects through enteric and respiratory routes, but causes little-to-no symptoms. Though quickly cleared by normal cells and healthy tissues, reovirus successfully replicates in transformed cells and tumors owing (in part) to damped host antiviral mechanisms [15–18]. We recently described that single amino acid differences between highly related reovirus T3D laboratory strains of a single ancestral human isolate, alter oncolytic potency *in vitro* and *in vivo* [19, 20]. Of 20 polymorphisms between T3D strains, 5 coding changes dispersed among 3 viral proteins played dominant effects on *in vitro* oncolytic activity. The most-oncolytic T3D$^{PL}$ strain displayed enhanced replication in a single round of infection, leading to higher burst size and enhanced cell death. Two mechanisms that contribute to the heightened replication of the most-oncolytic reovirus T3D$^{PL}$ strain relative to the least-oncolytic T3D$^{TD}$ were identified; T3D$^{PL}$ exhibited higher cell attachment and faster viral RNA transcription rates attributed to polymorphisms in the S1-encoded σ1 cell attachment protein and the M1-encoded μ2 NTPase, respectively. Overall, these findings indicated that T3D$^{PL}$, through small genomic divergence, had inherent and cell-response-independent advantages during replication in transformed cells.

In addition to inherent advantages in virus replication, it was possible that T3D strains differentially induced host responses and affected virus replication kinetics and oncolytic efficiencies. Indeed, previous studies found differences in interferon responses to distinct reovirus serotypes and lab strains [21–27]. However, these studies did not investigate the T3D$^{PL}$ strain which is currently in PhaseI/II cancer clinical trials nor compare strains with known differences in *in vivo* oncolytic activities. Moreover, previous studies did not decipher if differences in IFN signalling impinged on reovirus replication, and whether interferon induction was modulated by direct activities of reovirus proteins or indirect mechanisms. Accordingly, we set out to characterize the effects of naturally occurring genomic variations among T3D strains on host response to reovirus and were led to several unexpected discoveries that likely encompass on both therapeutic and pathogenic viruses.

Not surprisingly, given past differences between reovirus serotypes and strains, T3D lab strains induced different levels of RIG-I and/or IFN-dependent genes; specifically, the most-oncolytic T3D$^{PL}$ strain induced significantly lower levels than the least-oncolytic T3D$^{TD}$ strain. Unexpectedly, IFN production and response pathways did not affect the first round of T3D replication, suggesting that improved replication kinetics of T3D$^{PL}$ are independent of IFN antiviral effects. Commonly, virus mutations are predicted to affect host antiviral signalling by reducing activity of direct-acting viral antagonists of IFN signalling. In contrast to this conventional notion, we found that slow kinetics of virus replication was the cause of RIG-I/IFN-dependent host gene induction; specifically, replication of parental T3D$^{PL/TD}$ strains and reassortants inversely correlated with IFN signalling and drug-induced deceleration of T3D$^{PL}$ replication kinetics restored IFN signalling. Most surprising, however, was that during co-infection, the slow-replicating T3D$^{TD}$ strain phenodominated for IFN signalling over T3D$^{PL}$, indicating that slow replication produces a "trigger" that is not antagonized by the fast replicating strain. Together these findings suggest that similar to defective virions, small genetic

variations in infectious virions that slow replication rates contribute to heightened antiviral signalling. Finally, whole genome expression analysis and confirmatory RT-qPCR revealed that T3D$^{PL}$ induced higher expression of RIG-I/IFN-*in*dependent host chemokines and death-related proteins. Unlike the RIG-I/IFN-dependent signalling caused by slow rate of virus replication, induction of RIG-I/IFN-*in*dependent genes were attributed to polymorphisms in the dsRNA binding protein and PKR antagonist, σ3. This finding illustrates the sensitivity of virus-host relationships, and that small genetic differences between the same ancestral virus, in the same host, can lead to opposing cell signalling and cytokine landscapes.

## Results

### T3D$^{TD}$ activates interferon signalling more than T3D$^{PL}$

As described in the introduction, despite only few polymorphisms between reovirus T3D lab strains, the viruses exhibit gross differences in their oncolytic activity *in vivo* and single-step growth kinetics in a panel of cancer cells [19, 20]. Previous comparisons revealed that T3D lab strains possess inherent cell-independent differences, for example in the rate of transcription in a cell-free system. However, in addition to viral factors that impact virus replication, host interferon signalling can inhibit virus infection. We therefore wondered if T3D strains differ in the level of IFN induction, and whether antiviral signalling contributes to differences in virus growth kinetics between T3D strains. Our comparisons focused on the most-oncolytic and most-rapidly-replicating T3D$^{PL}$ strain versus the least-oncolytic and slowest-replicating T3D$^{TD}$ strain.

The antiviral response to viruses is very well characterized (Fig 1A). Upon entry into host cells, RNA viruses activate pathogen associated molecular pattern (PAMP)-recognition receptors (PRR) such as cytosolic retinoic acid-inducible gene I (RIG-I), melanoma differentiation-associated factor 5 (MDA5), laboratory of genetics and physiology 2 (LGP2), protein kinase R (PKR) and endosomal toll-like receptor 3 (TLR3). With reovirus, the dsRNA genome is encapsulated within a shell of core proteins throughout infection and presumed to be protected from PRRs. But within the core particle, dsRNA is transcribed into mRNAs that are released into the cytoplasm for translation by host machinery. It is proposed that reovirus PAMPs are generated from occasional unstable cores and/or viral mRNA secondary structures [28]. Activation of PAMP receptors, through a cascade of adaptor proteins and signalling events, results in phosphorylation (activation) and nuclear translocation of IRF3 and NFκB transcription factors, which subsequently induce expression of type I interferons and a subset of antiviral interferon stimulated genes (ISGs). Secreted IFNs bind to their cognate receptors, resulting in activation of the JAK-STAT signalling pathway in autocrine and paracrine fashion, and induction of many additional ISGs. ISGs can exert antiviral activities, for example ISGs cholesterol-25-hydroxylase and interferon-inducible transmembrane (IFITM) proteins were found to restrict reovirus [29, 30]. Given the possible impact of IFN production and response pathways on virus infection, an obvious question was whether T3D$^{TD}$ induces more robust IFN signalling, and if so, whether this contributes to the reduced replication of T3D$^{TD}$ relative to T3D$^{PL}$.

First, we determined if there were differences in IFN signalling between T3D laboratory strains. Phenotypes were first characterized in tumorigenic L929 mouse cells [31–33], since our previous comparisons found that enhanced oncolytic properties of T3D$^{PL}$ in L929 cells was representative of a panel of human and mouse cancer cell system [19, 20]. L929 cells were either mock-infected, or exposed to T3D$^{TD}$ or T3D$^{PL}$ at matched infectious doses of 1, 3, or 9. At 18hpi, immunocytochemical staining with anti-reovirus polyclonal antiserum confirmed that MOI of 9 resulted in near-complete infection of all cells while decreasing MOIs caused fewer cells to become positive for reovirus antigens (Fig 1B). The previously described delayed

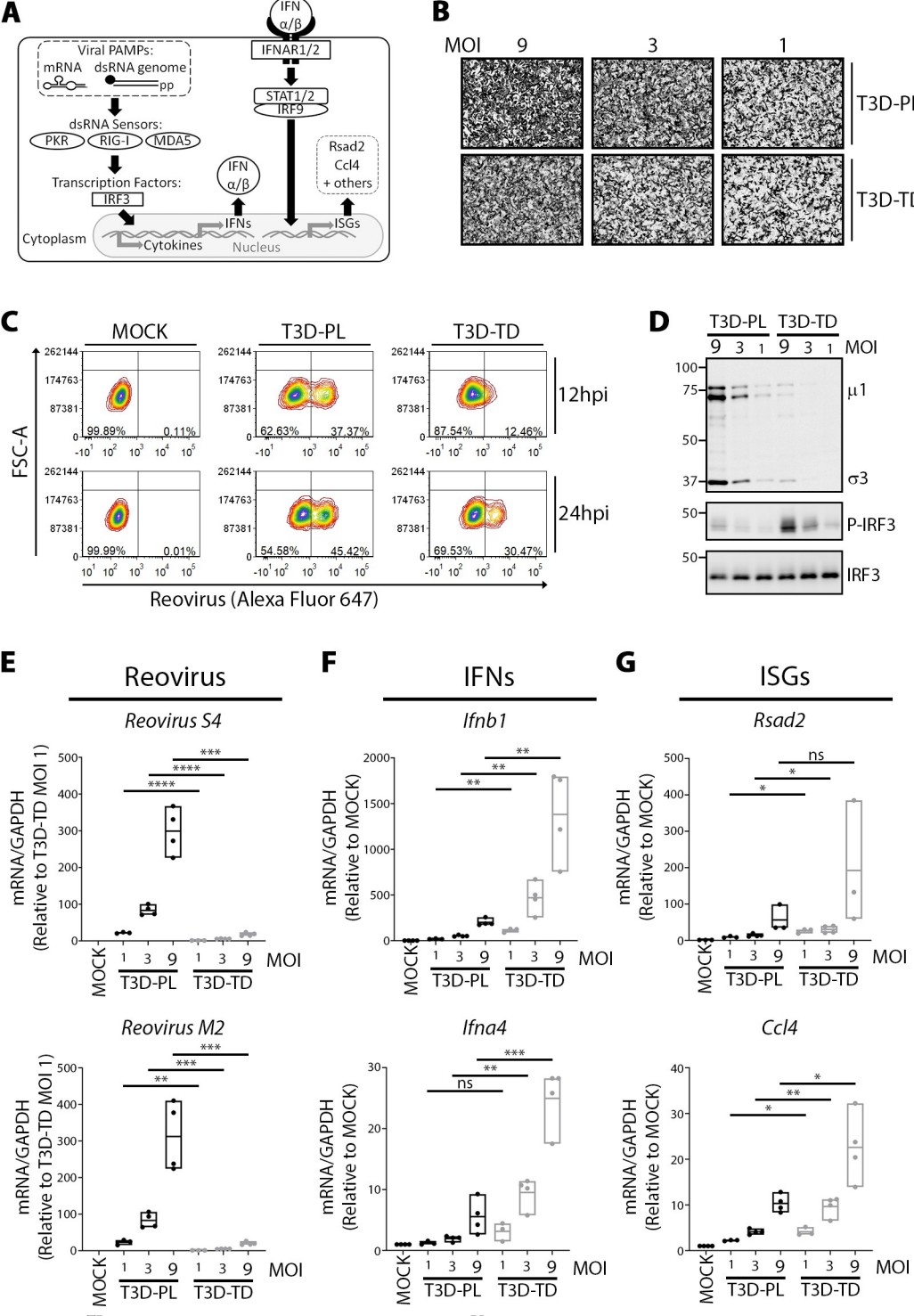

**Fig 1. T3D$^{TD}$ activates interferon signalling more than T3D$^{PL}$.** **(A)** Overview of IFN production (left) and response (right) antiviral signalling pathways. **(B-G)** L929 cells were infected with T3D$^{PL}$ or T3D$^{TD}$ at indicated MOI and incubated at 37˚C for 12 or 24 hours. **(B)** Immunocytochemical staining with anti-reovirus polyclonal antiserum shows cells positive for reovirus protein expression at respective MOIs. **(C)** To monitor the rate of establishing infection, flow cytometric analysis using reovirus polyclonal antiserum shows the proportion of L929 cells positive for reovirus antigen expression at 12 versus 24hpi. Mock-infected controls confirm that the left quadrant are cells with limited-to-no reovirus antigen detection while right quadrant are cells sufficiently infected to produce detectable reovirus antigen levels. **(D)** Total proteins collected at 12hpi were separated using SDS PAGE and Western blot analysis with antibodies for reovirus

proteins (σ3 and μ1), IRF3, or phosphor-IRF3. **(E-F)** At 12hpi, total RNA was extracted, converted to cDNA and gene expression relative to GAPDH was quantified by RT-qPCR. **(E)** Reovirus S4 and M2 reovirus RNAs, **(F)** type I IFNs *Ifnb1* and *Ifna4*, **(G)** ISGs *Rsad2 and Mx1*. Reovirus gene values were normalized to T3D-TD MOI 1 whereas cellular gene values were normalized to MOCK. Each point represents a biological replicate from n = 3–4 independent experiments. Statistical analysis represents unpaired t-tests comparing T3D$^{TD}$ and T3D$^{PL}$ of matched MOI. **** p ≤ 0.0001, *** p ≤ 0.001, ** p ≤ 0.01, * p ≤ 0.05, ns > 0.05.

kinetics of T3D$^{TD}$ [19, 20] is evident by quantifying the number of reovirus-antigen-positive cells by flow cytometric analysis at 12hpi (intermediate) versus 24hpi (late) stages of replication; T3D$^{TD}$ required 24 hours to achieve the level of infection seen by T3D$^{PL}$ at 12hpi (Fig 1C). Virus protein expression and IFN signalling were assessed at 12hpi, an intermediate time-point of virus replication when the potentially confounding effects of cell-cell virus spread are minimal. At 12hpi, T3D$^{PL}$ infection led to higher virus protein levels than T3D$^{TD}$ at the same infectious dose, consistent with enhanced replication kinetics (Fig 1D, μ1 and σ3 reovirus proteins). Strikingly, IRF3- phosphorylation was strongly induced by T3D$^{TD}$ relative to T3D$^{PL}$ (Fig 1D), despite the reciprocal trend for reovirus proteins.

The levels of virus versus IFN/ISG mRNAs were then compared for T3D strains using RT-qPCR. As anticipated because of the slow kinetics of T3D$^{TD}$ replication, viral mRNAs corresponding to S4 and M2 genome segments were lower for T3D$^{TD}$ relative to T3D$^{PL}$ (Fig 1E). While both T3D laboratory strains caused a dose-dependent increase in transcripts of IFNs *Ifnα4* and *Ifnβ*, T3D$^{TD}$ induced higher expression relative to T3D$^{PL}$ at equivalent dose (Fig 1F). Similarly, T3D$^{TD}$ induced higher expression of ISGs *Rsad2* and *Ccl4*, relative to T3D$^{PL}$ at equivalent dose (Fig 1G). Increased phosphorylation of IRF3 by T3D$^{TD}$ relative to T3D$^{PL}$ was reproduced in A549 human lung carcinoma cells (S1A–S1C Fig). Increased induction of IFN/ISGs by T3D$^{TD}$ was further confirmed in ID8 murine ovarian cancer cells, and in the B16-F10 murine melanoma system where T3D$^{PL}$ was previously characterized to exert more oncolytic activity *in vivo* relative to T3D$^{TD}$ (S1D and S1E Fig). Overall, despite producing lower levels of viral proteins and transcripts, T3D$^{TD}$ triggered greater IFN-production and IFN-response signalling pathways.

## T3D-strain specific polymorphisms in S4, M1, and L3 genes inversely affect IFN induction versus virus replication kinetics

We previously found that polymorphisms in S4, M1 and L3 genes were dominantly responsible for the increased replication of T3D$^{PL}$ in tumorigenic cells [19]. We wondered if S4, M1, and/or L3 were determinants of differential IFN production and response between the T3D strains. Mono-reassortants were generated by reverse genetics to exchange S4, M1, or L3 genome segments between T3D strains. Differences in plaque size, as an overarching measure of virus replication, release, and cell-cell dissemination, were confirmed for all viruses (Fig 2A). Mono-reassortants with T3D$^{PL}$ S4, M1 or L3 genes in an otherwise T3D$^{TD}$ parental background gave intermediate plaque size between T3D$^{PL}$ and T3D$^{TD}$ parental strains (Fig 2A, TD-RG Backbone). The same three genes also decreased plaque size when taken from T3D$^{TD}$ and reassorted into an otherwise T3D$^{PL}$ (Fig 2A, PL-RG Backbone). Having confirmed the phenotype of the mono-reassortants, we then measured induction of IFNs (*Ifnb1* and *Ifna4*) (Fig 2B) by RT-qPCR. Surprisingly, in an otherwise T3D$^{PL}$ genomic background, S4, M1 or L3 from T3D$^{TD}$ had limited effect on IFN induction. Conversely, when S4, M1 or L3 from T3D$^{PL}$ were individually reassorted into an otherwise T3D$^{TD}$ background, all three genes reduced IFN induction but not to the levels of T3D$^{PL}$. To explore the possibility that differential levels of defective virions in the virus preparations could impact IFN induction, we compared specific infectivity of the viruses. The ratio of total particles versus infectious particles was

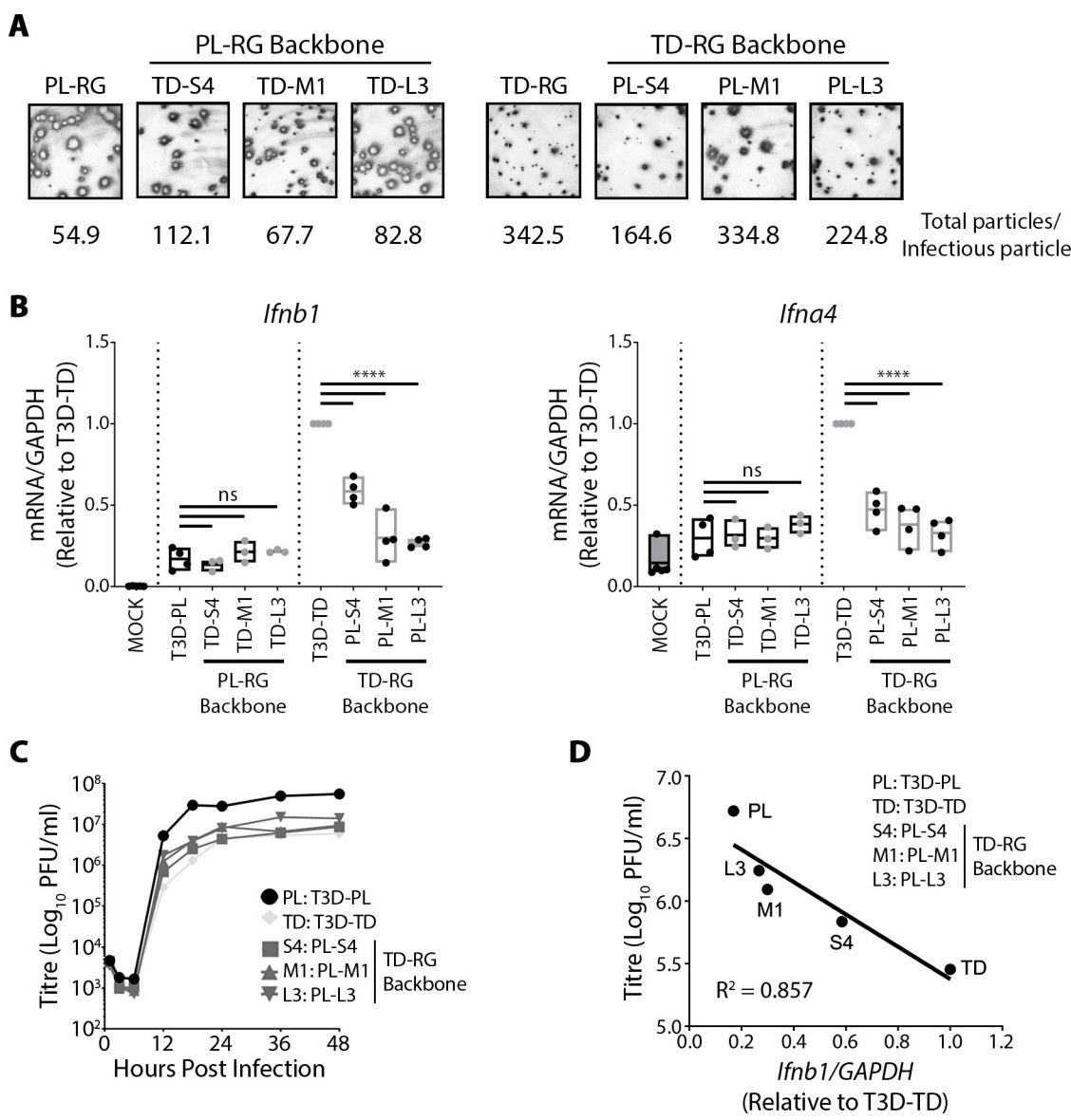

**Fig 2. T3D-strain specific polymorphisms in S4, M1, and L3 genes affect IFN induction inversely to virus replication kinetics.** (**A**) Reovirus foci generated by parental and mono-reassortant T3D^PL and T3D^TD viruses on L929 cell line. Reovirus foci were stained with colorimetric immunocytochemistry using primary polyclonal reovirus antibody, alkaline phosphatase secondary antibody and BCIP/NBT substrate. Total particles were determined using $OD_{260}$ of CsCl purified virus preparations ($1OD_{260} = 2.1 \times 10^{12}$ particles /ml). Infectious particles were calculated using standard plaque assays on L929 cells. (**B**) L929 cells were exposed to T3D^PL and T3D^TD parental viruses, or S4-, M1-, or L3- mono-reassortants in otherwise PL- or TD- backgrounds at MOI 3. At 12hpi, total RNA was extracted, and RT-qPCR performed for *Ifnb1* or *Ifna4*, each point represents a biological replicate from n = 3–4 independent experiments. One-way ANOVA with Dunnett's multiple comparisons test, **** $p \leq 0.0001$, ns > 0.05. (**C**) Single-step virus growth analysis was conducted by measuring total titers over time. Each point represents the average of 2 technical replicates. (**D**) Relationship of total viral titres versus *Ifnb1* gene expression (both at 12hpi) evaluated by linear regression analysis using PRISM; coefficient of determination ($R^2$) is provided. Average *Ifnb1* gene expression values obtained from Fig 2B.

determined using spectrophotometry at OD260 and standard plaque assay, respectively (Fig 2A, values below plaque photographs). There was no clear relationship between the specific infectivity and IFN induction (Fig 2A and 2B); therefore, increased IFN induction could not be easily explained by an increased proportion of defective or non-productive virions.

One possible interpretation of the data in Fig 2B, is that S4, M1, and L3 each independently contribute 'somewhat' to IFN signalling, such that mono-reassortants are insufficient to confer the full parental phenotype. Another possibility, however, is that these genes modulate virus replication kinetics and thereby indirectly, affect IFN induction. In support for the first possibility, other laboratories characterized reassortants between reovirus serotypes, serotype 1 Lange (T1L) versus serotype 3 Dearing (T3D), and found that S1, S2, S4, M1 and L2 reovirus genes associate with differences in IFN induction [22, 26, 27, 34–37]. Of these IFN modulating viral proteins, the S4-encoded σ3 was clearly demonstrated to sequester dsRNA, inhibit activation of PKR and rescue other viruses depleted of inhibitors of antiviral response [36, 38, 39]. But it should be noted that most studies on reovirus genes that impact IFN signalling do *not* consider whether effects are direct (e.g. the gene or protein directly modulate IFN mediators), or whether instead the viral genes impact virus replication and thereby indirectly impact IFN induction.

To consider virus replication kinetics as a possible contributor to IFN levels, we conducted single step growth curve analysis for parental and mono-reassortant viruses in the $T3D^{TD}$ parental background (Fig 2C). $T3D^{PL}$ exhibited faster single-step growth kinetics than $T3D^{TD}$ and reached higher burst size. At 12hpi and 18hpi, prior to saturation of viral titres, S4, M1, and L3 mono-reassortants presented intermediate growth kinetics relative to either parental strain. We then determined the relationship between IFN signalling (Fig 2B) and progeny titers (Fig 2C). A strong negative correlation ($R^2 = 0.857$) between virus replication proficiency and IFN signalling was found (Fig 2D). The correlation suggests that S4, M1, or L3 may contribute to differences in IFN signalling between $T3D^{TD}$ and $T3D^{PL}$ indirectly, by affecting the extent of virus replication.

## Deceleration of reovirus replication causes higher IFN induction

The inverse relationship between IFN production and replication kinetics for T3D parental and reassortant viruses suggested that rate of infection may be a key determinant of IFN induction for reovirus. To test this idea more directly, we reasoned that if true, then reducing the replication kinetics of the fast-replicating $T3D^{PL}$ strain should result in higher IRF3 phosphorylation. We therefore used guanidine hydrochloride (GuHCl) and cycloheximide (CHX) to impede virus replication of $T3D^{PL}$ and monitored IRF3 activation (Fig 3A). GuHCl was previously demonstrated to reduce reovirus dsRNA synthesis [40], and indeed caused a dose-dependent reduction of dsRNA corresponding to the Small (S), Medium (M) and Large (L) genome segments of reovirus (Fig 3B, bottom dsRNA electrophoresis). GuHCl also caused a dose-dependent reduction of reovirus proteins (Fig 3B, top Western blot analysis). CHX is a potent inhibitor of protein translation, and at all concentrations tested led to barely detectable levels of reovirus proteins and dsRNA (Fig 3B). Strikingly, as replication of $T3D^{PL}$ was reduced by either GuHCl or CHX treatment, IRF3 induction was increased in a dose-dependent fashion (Fig 3C). Even doses as little as 10-15mM GuHCl, which caused a visible but minimal reduction of $T3D^{PL}$ dsRNA and proteins, were sufficient to increase phosphor-IRF3 levels (Fig 3C and 3D). As important negative controls, in the absence of reovirus infection, CHX or GuHCl did not by themselves cause notable induction of IRF3 phosphorylation. Together, the GuHCl and CHX treatment data, along with data from T3D mono-reassortants, supports a model where slower kinetics of replication leads to heightened activation of the IFN production pathway.

## $T3D^{TD}$ IFN induction is not inhibited by $T3D^{PL}$ during co-infection

Two alternative explanations could account for increased IFN production and response by $T3D^{TD}$ despite less viral antigens relative to $T3D^{PL}$: (1) $T3D^{TD}$ might be more potent at

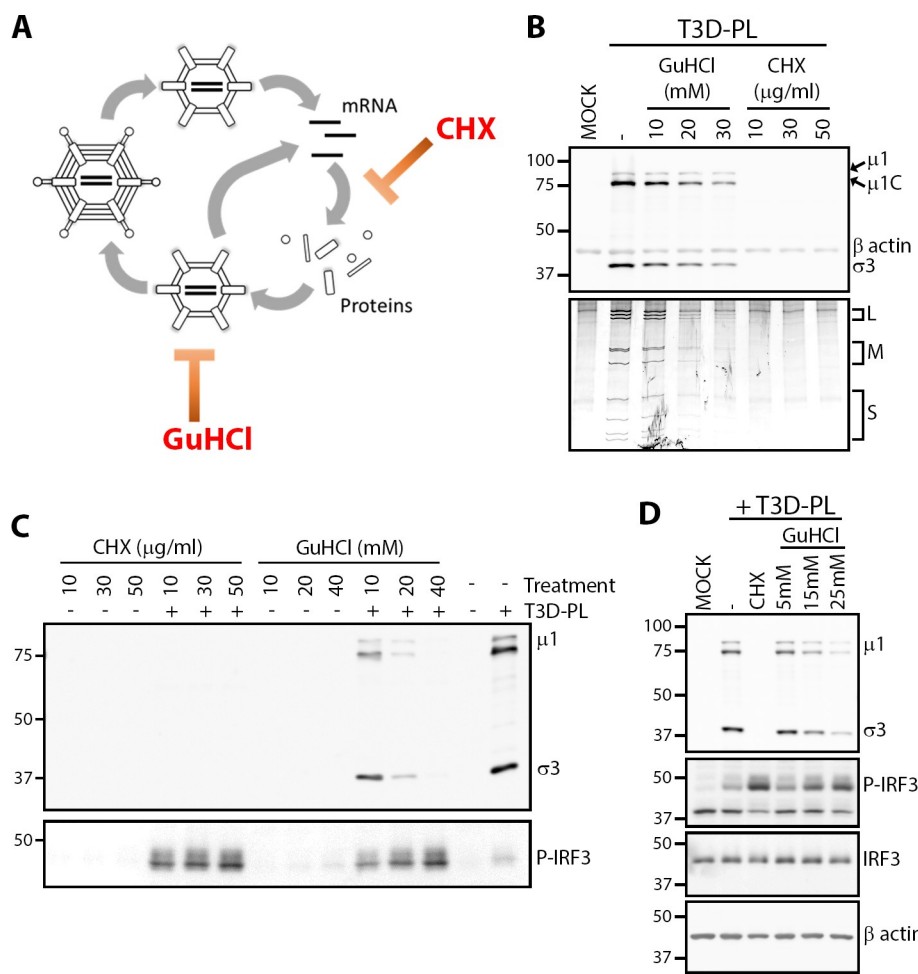

**Fig 3. Deceleration of reovirus replication causes higher IRF3 activation. (A)** Schematic of reovirus replication cycle with steps inhibited by cycloheximide (CHX) or guanidinium hydrochloride (GuHCl). **(B-D)** L929 cells were infected with T3D$^{PL}$ at MOI of 3 and treated with GuHCl or CHX through infection at indicated concentrations. At 12hpi, cells were subjected to **(B)** Western blot analysis for reovirus proteins and β-actin (Top) or dsRNA gel electrophoresis (Bottom). In independent experimental repeats, **(C)** cells were monitored by Western blot analysis for reovirus proteins and phosphor-IRF3, or **(D)** phosphor-IRF3, IRF3, and β-actin.

directly inducing antiviral signalling, or (2) T3D$^{PL}$ may be more potent at directly inhibiting antiviral signalling. To distinguish these possibilities, we reasoned that viruses containing a direct inhibitor of IFN production (e.g. a viral protein that sequesters IRF3) should be able to prevent IFN production during co-infection and rescue kin viruses devoid of the activity. Accordingly, L929 cells were co-infected with T3D$^{PL}$ and T3D$^{TD}$ at an MOI of 9 (each) to ensure that most cells were infected with both viruses, and cell lysates were subjected to Western blot analysis for IRF3 phosphorylation. Phospho-IRF3 levels were similar between T3D$^{TD}$ and T3D$^{PL}$/T3D$^{TD}$ co-infection (Fig 4A), suggesting that T3D$^{TD}$-dependant activation of IRF3 could not be overcome by the presence of T3D$^{PL}$. Furthermore, the levels of reovirus proteins were similar in T3D$^{PL}$/T3D$^{TD}$ co-infection relative to T3D$^{PL}$ despite high phospho-IRF3 levels, providing a first clue that IFN signalling may not play a major role in restricting the first round of reovirus replication; the role of IFN on reovirus infection will be revisited in subsequent figures.

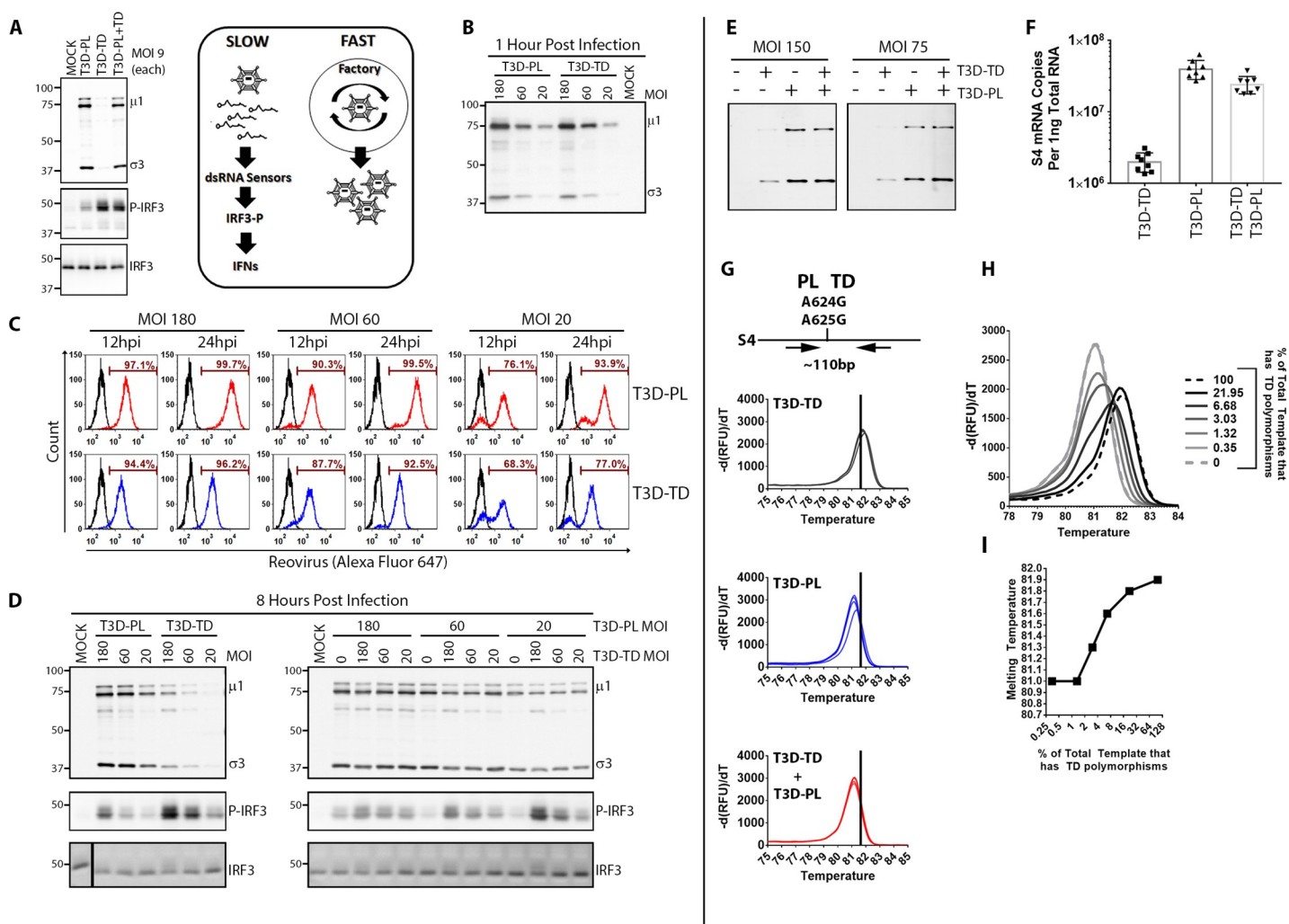

**Fig 4. T3D^TD is a potent inducer of IFN signalling that cannot be inhibited by T3D^PL during co-infection. (A, Left)** L929 cells were co-infected with T3D^PL and T3D^TD, each at MOI 9. At 12hpi, total proteins were separated by SDS PAGE and Western blot analysis performed with antibodies for reovirus proteins (σ3 and μ1), IRF3, or phosphor-IRF3. **(A, Right)** Model proposed from data in (A) and tested in (B-H), that during T3D^PL—T3D^TD co-infections, T3D^PL continues to replicate robustly while the slow-replicating T3D^TD continues to induce IFN signalling. **(B-D)** L929 cells were infected with T3D^PL and T3D^TD separately, or co-infected with both T3D strains, at MOIs indicated. Cells were first exposed to virus at 4°C to allow binding but not internalization, washed three times, and either (B) lysed for detection of cell-bound virions, or (C-D) transferred to 37°C for infection. **(B)** Cell-associated virions were visualized by Western blot analysis for reovirus proteins (σ3 and μ1), and demonstrated equivalent levels between T3D^PL and T3D^TD at matched MOIs. **(C)** Flow cytometric analysis with reovirus polyclonal antiserum at 12hpi and 24hpi showed the delayed infection by T3D^TD but that full infection of all cells was achieved at high MOI; this was to confirm that co-infection will occur at these MOIs when cells are exposed to both T3D strains. **(D)** Western blot analysis for reovirus proteins (σ3 and μ1), IRF3, or phosphor-IRF3 levels at 8hpi for individually (left) or co-infected (right) cells at indicated MOIs. **(E-F)** Western blot analysis **(E)** and RT-qPCR **(F)** show that during co-infection, high levels of reovirus protein and S4 RNAs phenodominate and resemble levels of T3D^PL–infected cells. Each point represents a biological replicate n = 8. **(G)** To determine if during co-infection, high viral protein and RNA levels were attributed to T3D^PL and/or T3D^TD replication, RT-qPCR was conducted with S4 HRM-specific primers and HRM analysis performed. Each plot represents two replicates each of cells exposed to the virus(es) indicated at MOI of 75 or 150. **(H)** qPCR and HRM was performed on cDNA from T3D^TD- infected cells, T3D^PL-infected cells, or mixtures of these two cDNAs in ratios back-calculated to produce the indicated percent of total RNA representing S4 polymorphisms from T3D^TD. The percentages were back-calculated from the RT-PCR levels obtained for T3D^TD-only versus T3D^PL-only samples. **(I)** The peak melting temperature from (H) plotted against the percent of S4 cDNA derived from T3D^TD.

The higher phosphorylated-IRF3 levels during co-infections suggested that T3D^TD is most likely a more potent activator of antiviral signalling than T3D^PL, as depicted in Fig 4A (right) as a model. Paradoxically, it suggested that the less-prolific replicating variant is dominant for triggering the IFN production pathway. This is paradoxical because less virus replication would normally be predicted to produce fewer PAMPs and therefore less IFN signalling. To

validate this unexpected finding, we therefore repeated co-infections at increasingly higher MOIs, and took extra steps to confirm that doses were more-than-sufficient to infect all cells. L929 cells were exposed to T3D$^{PL}$, T3D$^{TD}$, or both strains at MOIs of 20, 60, and 180. Viruses were permitted to bind to cells at 4°C for 1 hour, washed extensively, and then western blot analysis was performed to confirm that similar cell-bound virus particles were detected for T3D$^{PL}$ and T3D$^{TD}$ (Fig 4B). The infection was then allowed to proceed at 37°C for different timepoints. At 12hpi and 24hpi, the number of reovirus antigen-expressing cells was measured by flow cytometry (Fig 4C). As seen in Fig 1C, T3D$^{TD}$ infection was delayed relative to T3D$^{PL}$, but importantly, complete infection of all cells was achieved by both T3D strains. Levels of viral proteins and phosphor-IRF3 were then compared at 8hpi. There was a dose-dependent increase of phosphor-IRF3 with either T3D strain, but again, T3D$^{TD}$ caused higher IRF3 phosphorylation relative to T3D$^{PL}$ (Fig 4D, left panel). Importantly, regardless of the MOI, the levels of phosphor-IRF3 were determined by T3D$^{TD}$ dose (and not T3D$^{PL}$ dose) during T3D$^{PL}$-T3D$^{TD}$ co-infections (Fig 4D, right panel). Together, these experiments support that paradoxically, the slow-replicating T3D$^{TD}$ is a potent inducer of the IFN production pathway, which is not inhibited by T3D$^{PL}$ during co-infection.

During co-infections, high virus RNA and protein expression levels were detected, almost equivalent to T3D$^{PL}$ infection alone (Fig 4E and 4F). It was possible that high virus RNA and protein levels reflected continued rapid replication of T3D$^{PL}$ or could alternatively reflect that T3D$^{PL}$ rescued T3D$^{TD}$ replication and now both T3D strains were amplified to high levels. To distinguish these possibilities, High Resolution Melt (HRM) analysis was used to specifically quantify the levels of T3D$^{PL}$ versus T3D$^{TD}$ based on polymorphisms that existed in the reovirus S4 genome segment between the T3D strains. Specifically, oligonucleotide RT-qPCR primers were designed to flank the polymorphisms at nucleotide positions 624 and 625 of S4. The total amount of S4 copies was high for T3D$^{PL}$ and T3D$^{PL}$/T3D$^{TD}$ coinfected cells, relative to T3D$^{TD}$ infected cells (Fig 4F). The melting temperature, however, was lower for S4 amplicons derived from T3D$^{PL}$- versus T3D$^{TD}$- infected cells as would be predicted by the AA to GG transition mutation (Fig 4G). In co-infected cells, the melting temperature was similar to T3D$^{PL}$-derived S4, suggesting that at least the majority of S4 was from T3D$^{PL}$ origin. To determine the sensitivity of the HRM assay, we mixed cDNA from T3D$^{PL}$- versus T3D$^{TD}$- infected cells at known ratios and monitored melting temperatures of the qPCR S4 amplicons (Fig 4H and 4I). The S4 HRM assay was able to detect as little as 4% of T3D$^{TD}$ cDNA in a mixture of otherwise T3D$^{PL}$ cDNA. Altogether, it seems that during co-infections, T3D$^{PL}$- versus T3D$^{TD}$ incoming virions undertook separate fates, despite being in the same cell. T3D$^{TD}$ particles continued to replicate slowly and induced high IRF3 phosphorylation. T3D$^{PL}$ incoming particles continued to replicate at high kinetics and contributed minimally to IRF3 phosphorylation. That incoming virions can establish distinct fates in the same cell is surprising and will be further explored in the discussion.

## IFN signalling minimally restricts the first round of reovirus replication

Numerous studies have demonstrated the importance of the RIG-I/MDA5 signalling axis on reovirus mediated IFN production and subsequent paracrine suppression of reovirus spread to neighbouring cells [18, 41, 42]. However, whether autocrine RIG-I/MDA5 and IFN signalling can reduce the initial round of reovirus infection through autocrine activity is unknown. Given that T3D$^{TD}$ induced more IFN signalling than T3D$^{PL}$, the pivotal question became whether IFN signalling contributed to reduced replication kinetics of T3D$^{TD}$ relative to T3D$^{PL}$ in a single round of infection.

A direct and physiologically relevant approach to implicate a pathway in virus replication is to remove the pathway and determine if virus replication is changed. Accordingly, we made use of double knock-out (DKO) mouse embryo fibroblasts (MEFs) lacking both RIG-I and MDA5 [41, 43]. DKO cells were confirmed to show minimal induction of IFNs (*Ifnα4*) and IFN-induced gene *Rsad2* following 12 hours of infection by either T3D$^{PL}$ or T3D$^{TD}$ (Fig 5A). We reasoned that if IFN signalling can impact the first round of virus infection, then the DKO cells should demonstrate increased infection by reovirus at an intermediate timepoint of 12-18hpi where cell-cell spread is minimal. Wild-type (WT) and DKO MEFs were exposed to equivalent infectious dose of T3D$^{PL}$ or T3D$^{TD}$. The percent of cells productively expressing viral antigens was measured by flow cytometric analysis (Fig 5B) or visually monitored by fluorescence microscopy (Fig 5D, left). S4 reovirus mRNA production was quantified at 12hpi by RT-qPCR (Fig 5C). Virus input versus burst size were assessed by standard plaque titration (Fig 5D). By all three parameters of reovirus replication, when focusing on timepoints prior to cell-cell spread (≤18hpi), T3D$^{PL}$ outperformed T3D$^{TD}$ in the first round of infection as was anticipated. Most critically, knock-out of IFN production had minimal effects on either T3D strain during the initial round of replication. Note that IFN production did, however, exert the expected paracrine inhibitory effects on dissemination and subsequent rounds of reovirus replication, evident by increased numbers of infected cells (Fig 5B, 24hpi), titers (Fig 5D, 48hpi), and even plaque size (Fig 5E) in DKO relative to WT cells at time points ≥24hpi. The data was suggesting that T3D$^{TD}$ via slower replication kinetics causes greater IFN signalling, rather than greater IFN signalling causing slower T3D$^{TD}$ replication kinetics. Moreover, while differences in IFN signalling are not responsible for delayed replication kinetics of T3D$^{TD}$, they are likely to favor enhanced cell-cell spread for T3D$^{PL}$ over multiple rounds of infection.

The lack of autocrine effects of IFN on both T3D strains was surprising, and therefore was confirmed in a complementary system. The non-transformed NIH/3T3 mouse fibroblast cell line is well characterized to restrict reovirus via RIG-I dependent mechanisms [18]; though it was never determined if the restriction was autocrine or paracrine in nature. We therefore generated NIH3T3 cells that were stably silenced for RIG-I expression using shRNAs (shRIG) and compared these to NIH3T3 cells stably transduced with scrambled shRNAs (shSCR). RT-qPCR confirmed knock-down of RIG-I at mRNA (Fig 5F). Moreover, since RIG-I is also an ISG that is further upregulated by IFN signalling, it is not surprising that T3D$^{TD}$ induced ~2-fold higher levels of RIG-I than T3D$^{PL}$. RT-qPCR analysis also confirmed that IFNs (*Ifna4 and Ifnb1*) and ISGs (*Rsad2*, *Ccl4*, *Cxcl10*, *Ccl5*) were also expressed more by T3D$^{TD}$ than T3D$^{PL}$ in this cell line; but for either T3D strain, IFN and ISG expression was reduced when RIG-I was silenced (Fig 5G and 5H). Importantly, for either T3D strain, shSCR and shRIG-transduced cells showed equivalent levels of reovirus proteins by Western blot analysis (Fig 5I), and reovirus S4 and M2 mRNAs (Fig 5J) by RT-qPCR at 12hpi, indicating similar kinetics of viral de-novo macromolecular synthesis. Moreover, silencing of RIG-I did not change the number of reovirus antigen-positive cells in the first round of infection, as assessed by immunofluorescence staining (Fig 5K). The NIH3T3 system with shRNAs corroborates with the data from RIG-I/MDA5 DKO MEFs, suggesting that IFN induction has minimal restrictive effect on the first round of infection for T3D strains, and that IFN signalling does not account for the increased rate of infection by T3D$^{PL}$ relative to T3D$^{TD}$. The data suggests that T3D reovirus, at least in cell culture, secures one "free round" of replication impervious to the antiviral effects of IFN signalling.

As a third complementary strategy to assess the impact of IFNs on the first round of T3D reovirus replication, we treated L929 cells with IFNβ at various timepoints pre- and post- reovirus infection, and monitored cytokine expression (S1A Fig), virus titers (S1B Fig), or virus protein expression (S1C Fig) at 18hpi. We first validated that IFNβ treatment for 5hrs, in

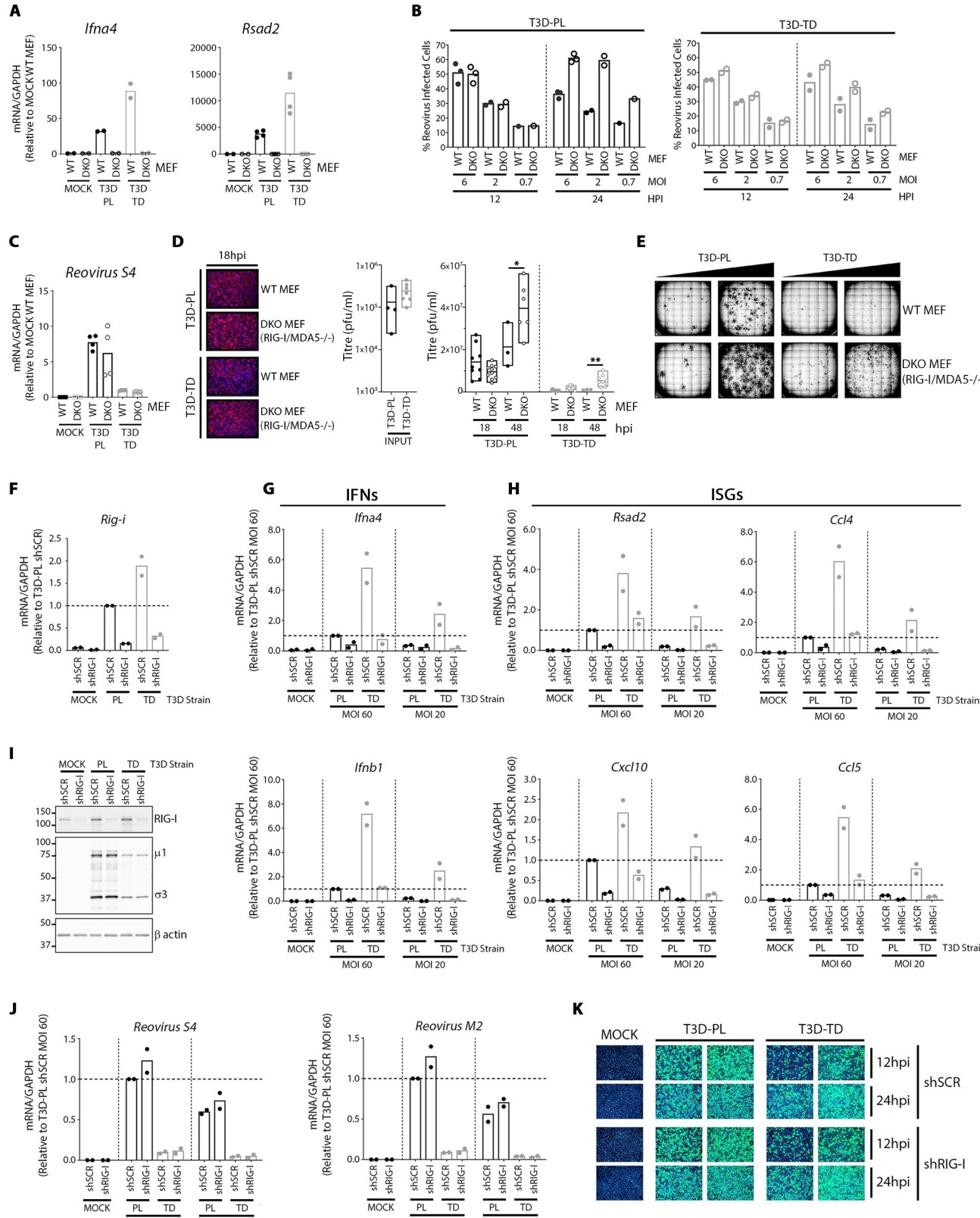

**Fig 5. IFN signalling minimally restricts the first round of reovirus replication. (A)** WT or RIG-I/MDA5 -/- double knockout (DKO) MEFs were infected with T3D$^{PL}$ or T3D$^{TD}$ at MOI 6 for 12hpi. Total RNA was extracted, converted to cDNA and gene expression relative to housekeeping gene

GAPDH was quantified using RT-qPCR for the genes indicated. All values were normalized to MOCK WT MEF. Each point represents a technical replicate for 1 (*Ifna4*) or 2 (*Rsad2*) independent experiments. (**B**) WT or DKO (RIG-I/MDA5 -/-) MEFs were infected with T3D$^{PL}$ or T3D$^{TD}$. At indicated MOIs and timepoints, percent of cells expressing reovirus proteins was determined by flow cytometry with polyclonal anti-reovirus antibodies and flow cytometry. Each point represents an independent experiment, n = 1–3 depending on MOI. (**C**) Same as (A) but RT-qPCR was conducted for reovirus S4 RNA levels, n = 2. All values were normalized to WT MEF T3D$^{TD}$. (**D**) WT or RIG-I/MDA5 -/- double knockout (DKO) MEFs were infected with T3D$^{PL}$ or T3D$^{TD}$ at MOI 1. (**Left**) Immunofluorescence staining at 18hpi confirms ~30% of cells are positive for reovirus antigen in all conditions. Reovirus specific primary antibody and Alexa Fluor 488 conjugated secondary antibody used to detect reovirus infected cells (Magenta), and HOESCHT 33342 to detect nuclei (Blue). (**Right**) Titres were determined by standard plaque assay on L929 cells for input, and progeny virus at 18 and 48hpi. (**E**) WT or DKO MEFs were infected with T3D$^{PL}$ or T3D$^{TD}$ under an agar overlay for 3 days. Reovirus infected cell foci were stained with colorimetric immunocytochemistry using primary polyclonal reovirus antibody, alkaline phosphatase secondary antibody and BCIP/NBT substrate. (**F-I**) NIH/3T3 cells stably transduced with scrambled (shSCR) or RIG-I (shRIG) lentivirus were mock-infected or infected with reovirus at MOIs of 20 and 60 (relative to titers obtained on the more-susceptible L929 cells), and incubated at 37˚C. (**F**) At 12hpi, RNA from cells infected at MOI of 20 was extracted and subjected to cDNA synthesis and RT-qPCR for mouse RIG-I (*Rig-i*) and mouse GAPDH. RIG-I mRNA levels were corrected for GAPDH and presented relative to shSCR T3D$^{PL}$ (set to 1.0). Each point represents a biological replicate n = 2. (**G and H**) Similar to (F) but RT-qPCR was conducted for (G) IFNs or (H) ISGs indicated above each plot, and following infection at MOIs of both 20 and 60 as indicated. Each point represents a biological replicate n = 2. (**I**) At 12hpi, samples were subjected to Western Blot analysis for RIG-I, reovirus µ1 and σ3, or β acting loading control. (**J**) Similar to (G) but RT-qPCR was conducted for viral genes indicated above each plot, and following infection at MOIs of both 20 and 60 as indicated. Each point represents a biological replicate n = 2. (**K**) At 12 and 24hpi, cells underwent immunofluorescence staining with reovirus specific primary antibody and Alexa Fluor 488 conjugated secondary antibody to detect reovirus infected cells (Green), and HOESCHT 33342 to detect nuclei (Blue). Green staining represents reovirus infected cells, and blue staining represents cell nuclei.

absence of virus, causes potent activation of prototypic ISGs (*Cxcl10*, *Rsad2*, *Mx1*), but not *non*-IFN-induced cytokines *Ifnb1* itself and *Cxcl1* (S2A Fig). In congruence with the paracrine effects of IFNβ on suppressing reovirus cell-cell spread in Fig 5, when cells were pre-treated with IFNβ for 5hrs prior to reovirus exposure (-5), there was strong inhibition of burst titers (S1B Fig) and protein synthesis (S1C Fig). To mimic potential autocrine effects of IFNβ on already-infected cells, recombinant IFNβ was added 3 or 5 hours post reovirus exposure. Effects of IFNβ on viral protein and titer accumulation was clearly diminished over time of infection, with minimal (not significant) effects by 3–5 hours post-infection for both T3D$^{PL}$ and T3D$^{TD}$. Therefore, addition of IFNβ at timepoints following reovirus entry and uncoating had nominal impact on virus replication.

Together the data supports the following model: T3D$^{TD}$ induces more IFN signalling than T3D$^{PL}$ because of slower replication kinetics, rather than that IFN signalling being the cause of reduced T3D$^{TD}$ replication kinetics. In other words, T3D$^{PL}$ inherently replicates better than T3D$^{TD}$ through mechanisms independent of IFN; some of these mechanisms were previously described and include inherently better transcription activity by T3D$^{PL}$ particles [19, 20]. However, since IFNs can suppress cell-cell spread, the induction of IFNs by T3D$^{TD}$ further suppresses dissemination of this T3D stain, producing an even-larger differential between T3D$^{TD}$ and T3D$^{PL}$ in IFN-proficient cells. As explored in the discussion, differential induction of IFN signalling by T3D$^{TD}$ and T3D$^{PL}$ are likely to affect immunological clearance of both virus and tumors and is therefore an important discovery.

## Reovirus also induces ~1500 IFN/RIG-I-independent genes

IFN/RIG-I-mediated innate responses are only one of many host signalling pathways activated during infection by viruses. For mammalian reovirus, previous studies found ERK and p38 signalling pathways positively associated with oncolytic activities [18, 44, 45], and NFκB signalling was necessary for induction of cell death and virus spread [46–48]. Given that T3D$^{PL}$ induced less IFN/RIG-I-dependent genes than other T3D lab strains, we grew curious to know if any IFN/RIG-I-independent genes were induced by T3D and whether their induction differs among strains. Whole-genome microarray analysis was conducted on three independent samples of NIH3T3 mouse fibroblasts infected by reovirus versus mock infected, and revealed 2357 host genes decreased by greater than 2-fold following infection, while 2947 increased by

greater than 2-fold (Fig 6A). We then generated NIH3T3 cells that stably expressed lentivirus-derived shRNAs to silence RIG-I (shRIG-I) or IFNAR (shIFNAR), or that stably expressed mouse RIG-I (mRIG-I O/E) to stimulate RIG-I dependent signalling (Fig 6B). NIH3T3 cells stably expressing an shRNA towards the irrelevant green fluorescence protein GFP (shGFP) were used as a negative control for possible confounding effects of transduction and puromycin selection. We reasoned that any host gene that is decreased by 2-fold in shRIG-I or shIFNAR cells relative to shGFP, or is increased 2-fold by mRIG-I O/E, is potentially a RIG-I- and/or IFN- dependent gene. The Venn diagram (Fig 6C) summarizes the number of genes that fell into each condition. Fig 6D diagrammatically depicts the role of each group during signalling, for example groups F, C, and D were affected by RIG-I silencing or over expression by not IFNAR silencing, suggesting they are downstream of RIG-I but not IFN. Strikingly, over half of T3D$^{PL}$-induced host genes were unaffected by all 3 manipulations of RIG-I or IFN, and therefore represent likely IFN/RIG-I-*in*dependent genes (group A).

With so many possible IFN/RIG-I-independent genes induced by T3D$^{PL}$ in NIH3T3 cells, it next became important to focus on host genes that were commonly induced in transformed cells. Whole-genome microarray analysis was repeated for L929 and B16-F10 mouse melanoma cells (Fig 6E). Since cells differ in susceptibility to reovirus, true MOIs of ~3 were used for each cell line to achieve approximately 70% infected cells. In general, NIH3T3 cells showed highest overall host gene expression in all groups A-H; whether this finding reflects a genuine difference between transformed versus non-transformed cells would need further investigation. Nevertheless, genes that were upregulated in all three cell lines during T3D$^{PL}$ infection emerged from this study and are summarized in Fig 6F. Among IFN/RIG-I dependent genes were many well-characterized ISGs such as *Rsad2* and *Mx1*, as well as IFNs such as *Ifnb1*. Most interesting were the genes in group A, representing IFN/RIG-I-independent genes such as *Fas*, *Tnfaip2*, *Nfkbie* and *Nfkbia* that impact cell death, and chemokines such as *Cxcl2*, *Cxcl1*, *Ccl2* and *Csf1* that could impact immunological responses to reovirus and oncolytic reovirus-targeted tumors.

## IFN/RIG-I-independent genes are induced differentially by T3D strains

Gene expression analysis revealed a subset of genes, including *Csf2*, *Cxcl1*, *Fas*, *and Cxcl2*, were upregulated during T3D$^{PL}$ infection independently of RIG-I and/or IFN activities. First, we sought to confirm the independence of these genes from RIG-I/IFN signalling. L929 cells were treated with IFNα or IFNβ and gene expression for IFN-dependent genes *Rsad2*, *Ccl4*, *Mx1*, *and Cxcl10* was confirmed by RT-qPCR (Fig 7A). In contrast, mRNA for *Csf2*, *Cxcl1*, *Fas*, *and Cxcl2* were not upregulated to high levels (>2 fold) by IFNα or IFNβ (Fig 7B). Moreover, NIH3T3 cells infected with T3D$^{PL}$ showed a stark upregulation of *Cxcl2*, *Csf2 and Fas*, genes that were insensitive to RIG-I shRNA-mediated silencing (Fig 7C). Surprisingly, *Cxcl1*, *Csf2*, *Cxcl2*, and *Fas* mRNAs were strongly induced (up to 30-fold) in a dose-dependent manner by T3D$^{PL}$ but not T3D$^{TD}$ infection (Fig 7C in NIH3T3, and Fig 7D in L929 cells). Most surprising about the findings, is that RIG-I/IFN-dependent versus independent signalling were inversely activated by T3D laboratory strains, which held true in five cell lines evaluated: NIH/3T3 (Fig 7C, n = 2), L929 (Fig 7D, n = 3), MEF (Figs 5A versus S1F, n = 2), B16-F10 (S1D Fig, n = 2) and ID8 (S1E Fig, n = 1). The data indicates that reovirus strains can differentially upregulate both IFN/RIG-I-dependent and independent genes, and that minor virus genomic diversity has potential to induce distinct host gene and cytokine profiles. As will be extrapolated in the discussion, the discovery of distinct cytokine expression landscapes between closely related T3D strains also suggests that virus genome manipulation may provide a new strategy to modulate the immunostimulatory potency of oncolytic viruses.

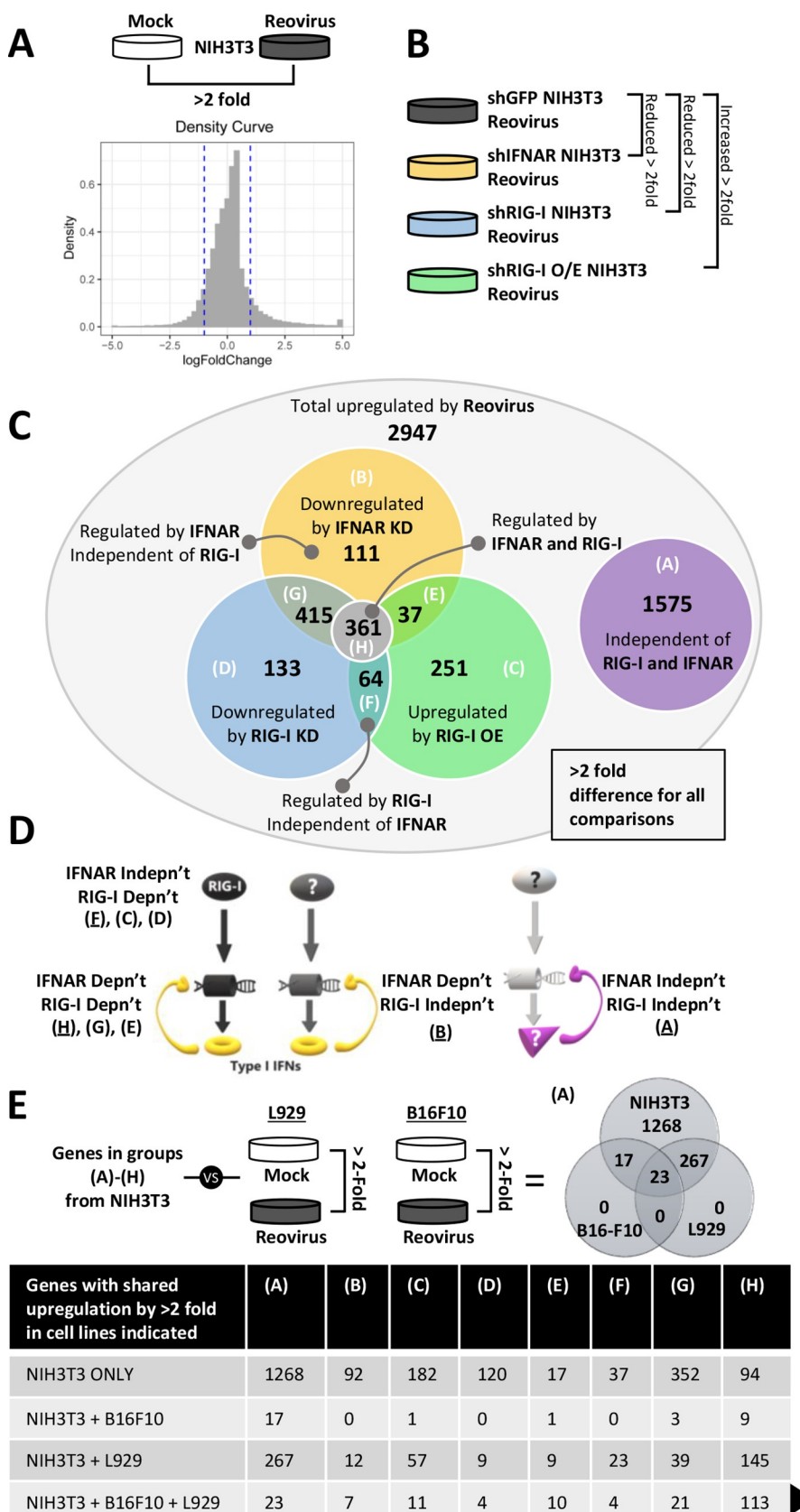

**(A) IFNAR Indepn't, RIG-I Indepn't (Top 10)**

| Gene | Fold change (Reovirus / Mock) | | |
|------|-------|------|--------|
| | NIH3T3 | L929 | B16F10 |
| Tnfaip3 | 53 | 3 | 9 |
| Mtmr7 | 51 | 5 | 12 |
| Cxcl2 | 48 | 12 | 36 |
| Csrnp1 | 48 | 3 | 5 |
| Cxcl1 | 37 | 4 | 14 |
| Fas | 23 | 3 | 31 |
| Atf3 | 15 | 3 | 2 |
| Nfkbie | 12 | 3 | 56 |
| Slc22a13b-ps | 10 | 2 | 6 |
| Relb | 9 | 2 | 4 |

**(B) IFNAR Depn't, RIG-I Indepn't**

| Gene | Fold change (Reovirus / Mock) | | |
|------|-------|------|--------|
| | NIH3T3 | L929 | B16F10 |
| Irf9 | 10 | 10 | 3 |
| C430049E01Rik | 7 | 2 | 14 |
| Gm1141 | 6 | 2 | 3 |
| Map3k8 | 6 | 3 | 4 |
| Lgals8 | 3 | 2 | 4 |
| A130078K24Rik | 2 | 2 | 2 |
| Shisa5 | 2 | 4 | 4 |

**(F) IFNAR Indepn't, RIG-I Depn't**

| Gene | Fold change (Reovirus / Mock) | | |
|------|-------|------|--------|
| | NIH3T3 | L929 | B16F10 |
| Ifnb1 | 3937 | 352 | 12 |
| Ccl5 | 643 | 61 | 8 |
| Ccl4 | 360 | 40 | 2 |
| Myd88 | 3 | 7 | 2 |

**(H) IFNAR Depn't, RIG-I Depn't (Top 10)**

| Gene | Fold change (Reovirus / Mock) | | |
|------|-------|------|--------|
| | NIH3T3 | L929 | B16F10 |
| Rsad2 | 13588 | 20 | 466 |
| Mx1 | 4546 | 37 | 910 |
| Iigp1 | 2342 | 9 | 186 |
| Mx2 | 1575 | 24 | 118 |
| Cxcl10 | 1525 | 18 | 178 |
| Gm4951 | 1220 | 5 | 50 |
| Oas2 | 1175 | 20 | 43 |
| Ifi44 | 1119 | 29 | 46 |
| Oasl1 | 735 | 28 | 19 |
| Cxcl9 | 652 | 8 | 213 |

**E**

| Genes with shared upregulation by >2 fold in cell lines indicated | (A) | (B) | (C) | (D) | (E) | (F) | (G) | (H) |
|------|-----|-----|-----|-----|-----|-----|-----|-----|
| NIH3T3 ONLY | 1268 | 92 | 182 | 120 | 17 | 37 | 352 | 94 |
| NIH3T3 + B16F10 | 17 | 0 | 1 | 0 | 1 | 0 | 3 | 9 |
| NIH3T3 + L929 | 267 | 12 | 57 | 9 | 9 | 23 | 39 | 145 |
| NIH3T3 + B16F10 + L929 | 23 | 7 | 11 | 4 | 10 | 4 | 21 | 113 |

**Fig 6. Reovirus also induces RIG-I/IFN-independent genes. (A)** Density plot shows overall number of genes up and down regulated in whole genome microarray analysis conducted on NIH3T3 cells treated with Reovirus versus mock infected. **(B)** Experimental approach for (C-D): Whole genome microarray analysis was conducted for reovirus-infected NIH3T3 cells stably transduced by lentivirus expressing small hairpin RNAs to silence IFN receptor (shIFNAR), RIG-I (shRIG-I), retrovirus that overexpresses mouse RIG-I (mRIG-I O/E), or negative control shRNAs against green fluorescence protein. The diagram shows the comparison groups and parameters used for subsequent analysis in (C). **(C)** Venn diagram of all 2947 genes upregulated by reovirus in (A), grouped into whether they were downregulated by IFNAR silencing (yellow), downregulated by RIG-I silencing (blue), or upregulated by RIG-I over-expression (green), or unaffected by RIG-I and IFNAR modulation (purple). Each group and overlap in the Venn Diagram was then assigned a letter (A-F) to reflect that group. **(D)** Diagram shows where groups A-F fit within signalling pathways that produce IFNs (yellow) versus IFN-independent (purple) genes. **(E)** Whole genome microarray analysis was conduced for reovirus infected versus mock infected L929 cells and B16-F10 cells. The table summarizes the number of genes in groups A-F that were also upregulated by 2-fold in L929 and B16-F10 cells, as well as those shared by all three cell lines. **(F)** Tables show the genes in key groups used for subsequent analysis (A, B, F and H) shared among NIH3T3, L929 and B16-F10 cells. For group H, only the top 10 genes are listed due to space constraints. Complete lists of all genes in each group are provided in S1 Table.

## T3D$^{PL}$ S4-encoded σ3 stimulates expression of RIG-I/IFN-independent cytokines

Having discovered that T3D$^{PL}$ induces RIG-I/IFN-independent genes more than T3D$^{TD}$, we returned to the S4, M1, and L3 monoreassortants to determine if any of these genes contributes to differential induction of *Cxcl1*, *Csf2*, *Cxcl2*, *and Fas*. All four host mRNAs were upregulated by T3D$^{PL}$ more than T3D$^{TD}$ (Fig 8), in agreement to results from Fig 7. Importantly, greater *Cxcl1*, *Csf2*, *Cxcl2*, *and Fas* expression strongly corresponded with polymorphisms in the S4 genome segment encoding the σ3 protein. When the T3D$^{TD}$-derived S4 was introduced into an otherwise T3D$^{PL}$ parental genome background, expression of these RIG-I/IFN-independent genes reduced to levels of T3D$^{TD}$. When T3D$^{PL}$-derived S4 was introduced into an otherwise T3D$^{TD}$ parental genome background, *Cxcl1*, *Csf2*, *Cxcl2*, *and Fas* mRNA levels increased but not to the level of T3D$^{PL}$; most likely because expression of all reovirus proteins including σ3 are reduced in a T3D$^{TD}$ background with slow replication kinetics [20]. The σ3 protein has two well characterized activities; it functions as an outer-capsid protein and it sequesters viral RNAs away from cellular dsRNA-detecting signalling molecules such as PKR and RIG-I [38, 39]. Our data now suggests that σ3 contributes to expression of RIG-I/IFN-independent host responses. Altogether, the data indicates that while T3D$^{TD}$ induces RIG-I/IFN -dependent host genes and cytokines owing to slow replication kinetics, T3D$^{PL}$ induces RIG-I/IFN-independent host genes and cytokines owing to polymorphisms in S4-encoded σ3. Only three polymorphisms exist between σ3 of T3D$^{PL}$ and T3D$^{TD}$, demonstrating a remarkable sensitivity of host signalling to small amino acid virus divergences.

## Discussion

T3D$^{PL}$ and T3D$^{TD}$ are highly related strains of reovirus that originate from the same isolate of serotype 3 Dearing. Over the course of propagation in different laboratories, these strains have accumulated 20 amino acid polymorphisms. Surprisingly, despite only small genomic divergence, the T3D lab strains exhibit drastic differences in their ability to replicate in tumorigenic cells *in vitro*, and exhibit oncolytic activities *in vivo* [19, 20]. Classical and reverse genetics approaches previously revealed that five polymorphisms dispersed among S4, M1, and L3 reovirus genes account for the oncolytic differences between T3D strains. Furthermore, it was found that the closely related T3D laboratory strains had cell-independent differences in replication kinetics, owing to intrinsically greater cell binding and transcription by the most-oncolytic T3D$^{PL}$ strain [19, 20]. However, given that cell signalling and host gene expression also affect both virus replication and oncolysis, it was important to characterize if cell signalling differed among the T3D strains.

The data in this manuscript suggests that despite being closely related strains of T3D, T3D$^{PL}$ and T3D$^{TD}$ induce opposite expression of RIG-I/IFN-dependent versus independent gene expression (Fig 9, Model). RIG-I/IFN-dependent signalling is reciprocal to replication

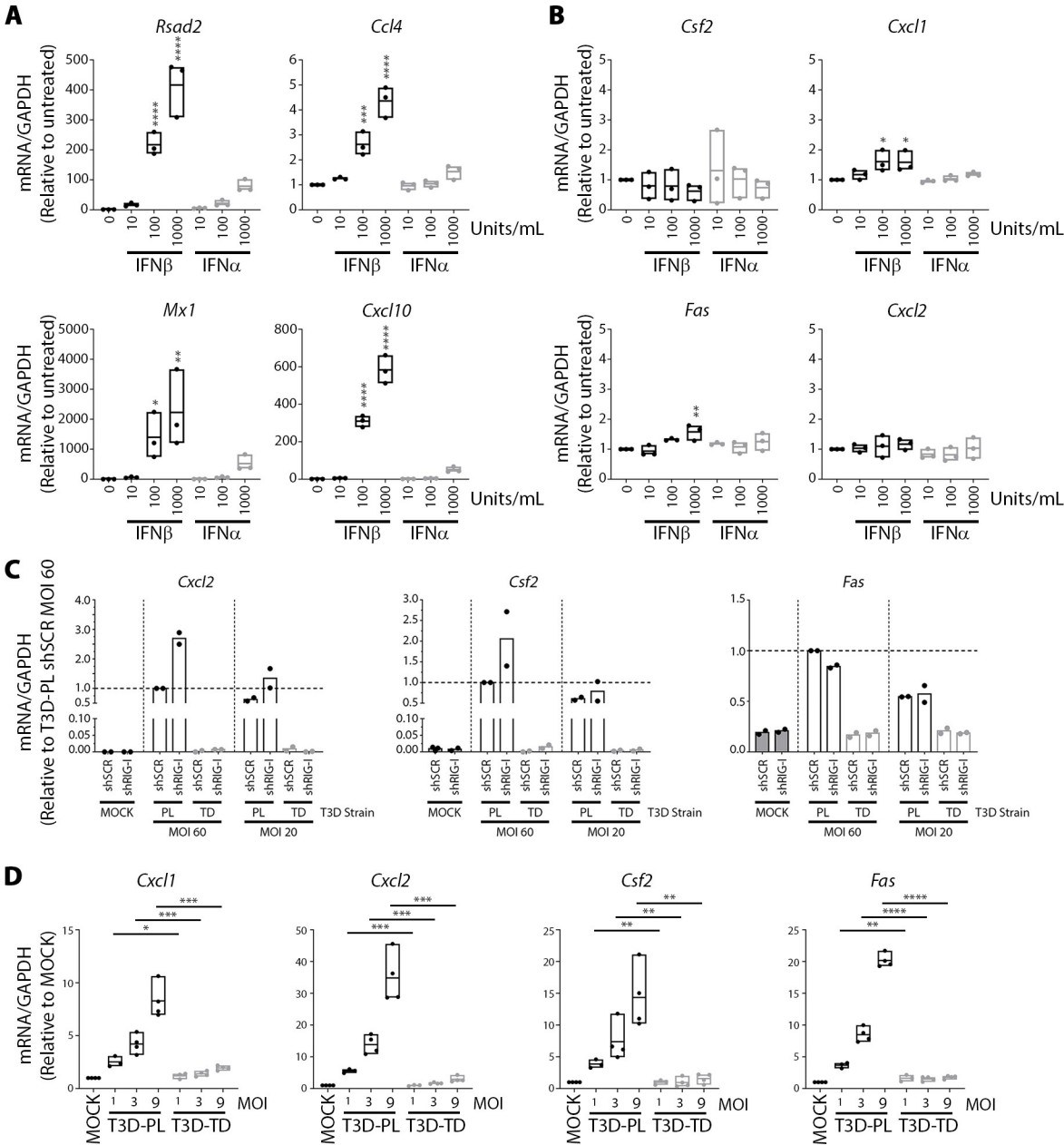

**Fig 7. RIG-I/IFN-independent genes are induced by differentially by T3D strains.** (**A-B**) L929 cells were treated with 1/10 dilutions (initial 1000 U/ml/12well) of purified IFNα or IFNβ for 12 hours at 37˚C. Samples were collected for RNA extraction, cDNA synthesis and RT-PCR using gene-specific primers (corrected for GAPDH) for (**A**) ISGs or (**B**) genes proposed to be IFN-independent from Cluster 6 of Fig 6. Values were standardized to untreated sample. Each point represents a biological replicate n = 3. One-way ANOVA with Dunnett's multiple comparisons test, **** $p \le 0.0001$, *** $p \le 0.001$, ** $p \le 0.01$, * $p \le 0.05$. (**C**) NIH/3T3 cells stably transduced with scrambled (shSCR) or RIG-I (shRIG) lentivirus were mock-infected or infected with reovirus at MOIs of 20 and 60 (relative to titers obtained on the more-susceptible L929 cells), and incubated at 37˚C. At 12hpi, RNA from cells infected at MOI of 20 was extracted and subjected to cDNA synthesis and RT-qPCR for *Cxcl2*, *Csf2*, *Fas* and mouse *GAPDH*. RIG-I mRNA levels were corrected for GAPDH and presented relative to shSCR T3D^PL (set to 1.0). Each point represents a biological replicate n = 2. (**D**) RT-qPCR shows levels of indicated genes following infection of L929 cells with T3D^PL or T3D^TD at MOI 1, 3, or 9. For each gene, levels are standardized to MOCK, which is set to 1. Each point represents a biological replicate n = 3–4. Statistical analysis represents unpaired t-tests comparing T3D^TD and T3D^PL of similar MOI. **** $p \le 0.0001$, *** $p \le 0.001$, ** $p \le 0.01$, * $p \le 0.05$.

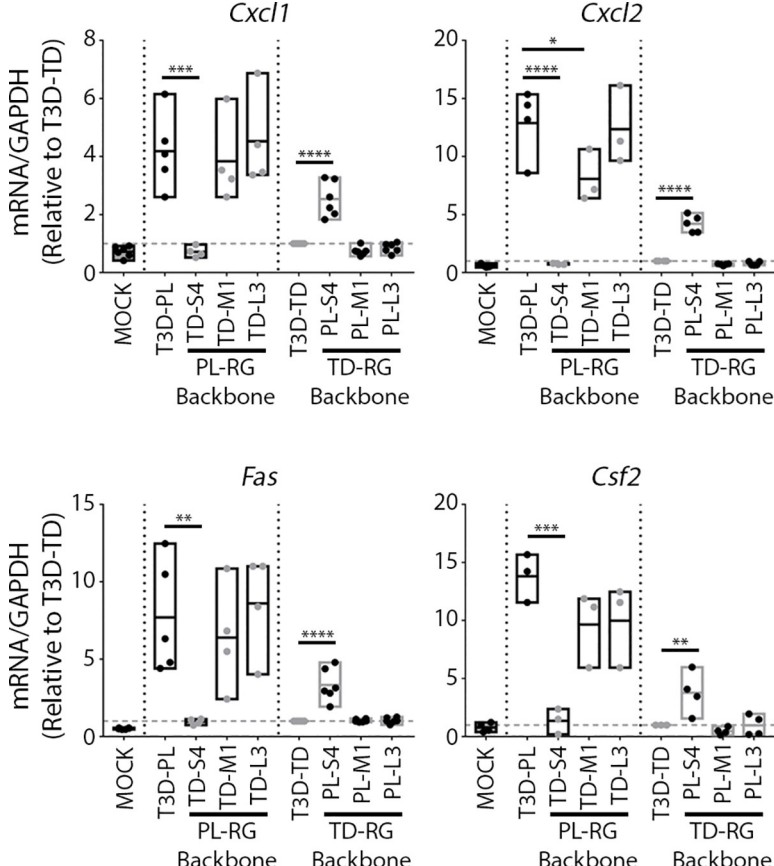

**Fig 8. T3D^PL S4-encoded σ3 stimulates expression of RIG-I/IFN-independent cytokines.** Standardized for equal infection (MOI 3), L929 cells were infected for 12hrs with parental T3D^PL or T3D^TD, and S4, M1 and L3 gene mono-reassortant TD-RG viruses. Total RNA was assessed for specified genes relative to housekeeping gene GAPDH. n = 3–4. Statistical significance determined using one-way ANOVA with Dunnett's multiple comparisons test, **** p ≤ 0.0001, *** p ≤ 0.001, ** p ≤ 0.01, * p ≤ 0.05.

kinetics, suggesting that slow-replicating reovirions activate more innate signalling. Paradoxically, the slow-replicating T3D^TD phenodominantly induced IRF3 activation during co-infection with fast-replicating T3D^PL, despite that T3D^PL phenodominantly produced high levels of virus proteins. This finding suggests that incoming reovirions with genomic diversity can establish distinct fates even within the same cell. The first round of reovirus replication is impervious to RIG-I/IFN mediated innate signalling, but subsequent dissemination of T3D^PL is enhanced by reduced activation of this pathway. Remarkably, host genes that were independent of RIG-I or IFN signalling were induced more by T3D^PL than T3D^TD and determined by polymorphisms in σ3. Together, the findings reveal how single polymorphisms in T3D grossly affect host response.

## Closely related strains can induce opposite cytokine landscapes

We find it remarkable that highly related, laboratory divergent viruses with only few polymorphisms can cause complete-opposite cellular signalling and gene expression outcomes. T3D^TD caused upregulation of RIG-I/IFN-induced cytokines such as CCL5, CCL4, and CXCL10, while conversely T3D^PL caused upregulation of RIG-I/IFN-*in*dependent cytokines such as CXCL1, CXCL2 and CSF2. It would be interesting to know if similar examples exist for other

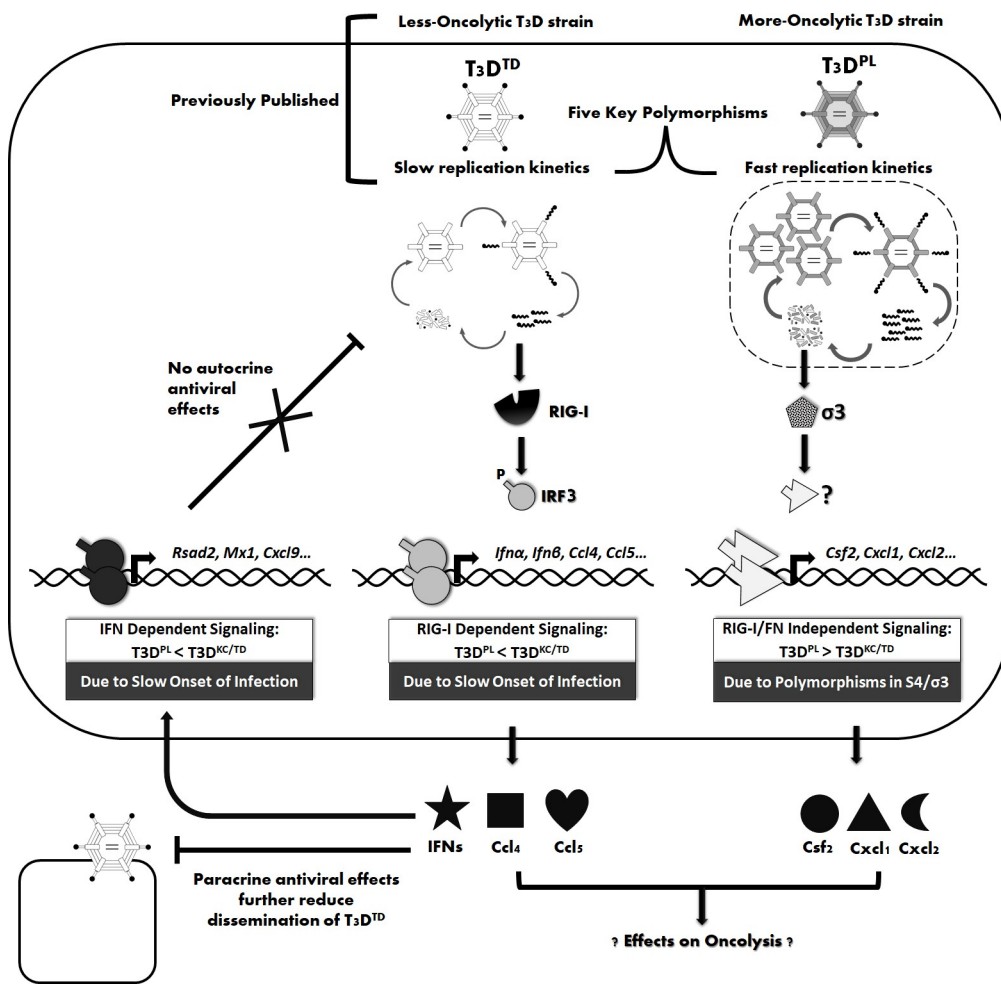

**Fig 9. Model: T3D<sup>PL</sup> versus T3D<sup>TD</sup> induce opposite expression of RIG/IFN-dependent versus independent gene expression; attributed to slow replication versus polymorphisms in σ3 respectively.** Previous studies described in text found that despite only 5 key polymorphisms, T3D strains had inherent differences in replication kinetics and in vivo oncolytic activities; with T3D<sup>TD</sup> being less oncolytic and slower replicating than T3D<sup>PL</sup>. In this manuscript, we discover that **(Left)** T3D<sup>TD</sup>, by virtue of slow replication kinetics, triggers more RIG-I-dependent IFN signalling relative to T3D<sup>PL</sup>. **(Right)** T3D<sup>PL</sup>, by virtue of rapid replication, avoids triggering of IFN producing and response pathways. Conversely, the polymorphisms in σ3 of T3D<sup>PL</sup>, lead to strong activation of RIG-I/IFN-independent genes. In the model, we depict a possible explanation for why T3D<sup>PL</sup> triggers less RIG-I/IFN signalling; by establishing sufficient segregation in virus factories (dotted line around replication complex), T3D<sup>PL</sup> overcomes PAMP detection and IFN signalling induction. Moreover, the results show that IFN does not impede T3D replication through autocrine antiviral action (i.e. ISG induction is not the reason for slow replication of T3D<sup>TD</sup>). However, antiviral signalling can further prevent dissemination of reovirus and could thereby contribute to reduced oncolytic activity of T3D<sup>TD</sup>. Finally, the model depicts that differential cytokine landscapes induced by distinct T3D strains could further contribute to differences in oncolytic activities in immunocompetent hosts; this new concept beacons further analysis on the effects of virus genomes on their immunotherapeutic potential.

viruses, and to what extent virus genomic divergence contributes to cytokine landscapes and pathogenesis. For example, would a quasispecies of virus, by representing a range of polymorphisms, cause a wider array of cytokines than a single clone? If so, would that be desirable for the virus, the host, or neither?

Reovirus-mediated oncolysis is a combination of both direct virus-induced cytolysis [49–53] and anti-tumor immune stimulation [54–59]. Given that cytokines, chemokines, and innate signalling molecules strongly influence anti-tumor immunity [60–64], we predict that

they will impact anti-tumor immunity during reovirus oncolysis and contribute to the immunotherapeutic value of reovirus. Our data provokes a close investigation into the roles of RIG-I/IFN-dependent versus independent cytokines on reovirus-strain-specific anti-tumor activity. Based on the 5 cytokines we analyzed, it is impossible to predict the impact of the distinct cytokine profile induced by each strain. As an example, CXCL10 is proposed to have both pro- and anti-proliferative action on breast cancer depending on the receptors it engages [65]. Most importantly, many more IFN-dependent and -independent cytokines are likely upregulated differentially by reovirus strains that were not evaluated in this study. A systematic approach will be necessary to precisely determine the benefits or costs of using reovirus strains that induce RIG-I/IFN-dependent versus independent cytokines. Should both groups of cytokines prove important during oncolytic virus treatment, then combination therapy with both T3D$^{PL}$ and T3D$^{TD}$ reovirus strains (or a hybrid variant that induces both types of signalling) warrants consideration. One must also consider that such cytokines would contribute to heightened clearance of the virus, and therefore to achieve a balance between strong virus replication and strong anti-tumor activity, immunologically distinct oncolytic viruses may need to be sequentially used. Overall, our findings suggest that small genomic differences among virus strains could impact the immunotherapeutic potential of an oncolytic virus.

## Slow replication can promote IFN induction, and phenodominate in co-infections

Understanding the mechanisms of IFN signalling induced by viruses is important for both virus oncolysis and virus pathogenesis. With respect to RIG-I/IFN-dependent genes, countless examples show that many virus encoded genes counteract cellular signalling. Indeed, even for reovirus, previous reassortant analysis between reovirus serotypes 1 Lang (T1L) and 3 Dearing (T3D) suggested that S2, S4, M1 and L2 reovirus genes function to regulate IFN signalling [26, 27, 34–37]. But our studies suggest an alternate mechanism affects activation of RIG-I/IFN-dependent genes; replication of mono-reassortant and parental viruses was inversely correlated with IFN induction and slowing T3D$^{PL}$ kinetics caused increased IRF3 phosphorylation. We therefore propose the following model: Slow onset of replication leads to accumulation of viral PAMPS, detection by RIG-I and MDA5, and activation of IFN production and response pathways. Reciprocally, fast replication results in rapid isolation of virus PAMPs in virus factories (i.e. localized areas of virus replication). In other words, if incoming reovirus cores fail to establish infection quickly, there is less protection from cellular proteins. That virus replication can inversely correlate with IFN induction is a timely finding, as there is growing interest in the concept that defective virions in a virus quasispecies contribute to innate detection of viruses [11, 66]. If our model is correct, then similar to T3D$^{TD}$, other virus mutants in a quasispecies with slow kinetics of establishing infection, even if not defective, may be inherently more vulnerable to innate detection. Slow replicating mutants could thereby help protect the host from rapidly replicating mutants by inducing antiviral signalling. Inversely, since hyperstimulation of innate signalling can also cause immune-associated pathologies [67, 68], slow-replicating mutants could instead exasperate pathological responses to viruses. These possibilities would be interesting to test directly, and with various RNA virus families.

Paradoxically, slow-replicating reovirus T3D$^{TD}$ phenodominated a strong antiviral IFN response, even when co-infected with fast-replicating T3D$^{PL}$. This is paradoxical because fast-replicating viruses are expected to produce more PAMPs that trigger IFN signalling and more viral inhibitors of IFN signalling; in either scenario producing a dominant outcome on IFN signalling. It is also unexpected that robust replication of T3D$^{PL}$ during co-infections failed to prevent IRF3 phosphorylation by T3D$^{TD}$ in co-infected cells; this suggests it is unlikely that

T3D$^{PL}$ globally counteracts RIG-I signalling. Together, our observations support that incoming virus particles during co-infection might behave independently, which is compatible with previous findings that virus factories are established around the incoming reovirus core [69, 70]. During co-infections, we predict that incoming T3D$^{PL}$ and T3D$^{TD}$ cores each initially establish their own factories. This is congruent with our findings that during co-infections, T3D$^{PL}$ incoming particles continue to replicate with fast kinetics and produce the dominant population of viral RNAs, while T3D$^{TD}$ retains slow replication without benefit from T3D$^{PL}$ in the same cell. We now propose that incoming cores must establish factories with sufficient speed to prevent detection of PAMPs by the host. Indeed, recent studies showed the ability of reovirus factories to selectively sort IFN signalling components [71]. In this scenario, the slow-replicating T3D$^{TD}$ virus would still cause induction of RIG-I/IFN signalling, by mere absence of rapid isolation in factories. But since IFN signalling does not affect the first round of reovirus replication (discussed next), induction of IFN by T3D$^{TD}$ would not impede the replication of T3D$^{PL}$, thus explaining the continued high levels of T3D$^{PL}$ proteins during T3D$^{PL}$/T3D$^{TD}$ co-infections. It should be noted however, that virus factories/inclusion bodies eventually converge into a large perinuclear body; since we and others can easily generate reassortants during co-infection, the overall model needs to consider that incoming strains, although exerting distinct effects on IFN induction early, eventually mix and exchange segments. The results beckon a better understanding of how incoming cores establish factories, and direct experiments to implicate factory formation kinetics in RIG-I/IFN-dependent signalling.

## RIG-I/IFN antiviral signalling does not restrict the first round of reovirus T3D replication

Another surprising result was that although IFN signalling clearly affected the plaque size of both T3D strains, suggesting paracrine effects on cell-cell spread, it did not affect the initial round of infection. Specifically, knock-out or knock-down of RIG-I was sufficient to eliminate IRF3 phosphorylation and expression of IFNs and ISGs during reovirus infection yet did not decrease the first round of replication for either T3D strains. These findings demonstrate that differences in replication between the T3D strains are not a consequence of IFN signalling. Our previous studies showed that T3D$^{PL}$ particles inherently exhibited higher binding and faster transcription rates than T3D$^{TD}$, which helps explain the different rates of infection in absence of effects by RIGI-I/IFN antiviral signalling [19, 20].

Studies on IFN signalling rarely distinguish between autocrine and paracrine effects, and this is the first time to our knowledge that IFN effects on reovirus are demonstrated to be futile during the first round of replication. If applicable to a natural infection, then reovirus would be afforded one "free round" of replication that is unrestricted by antiviral effects of IFN and may contribute to successful dissemination of reovirus among most humans and many mammals. Some possible explanations for the absence of autocrine effects are that perhaps ISG do not reach sufficient stochiometric levels in time to stop reovirus, or perhaps the ISGs effective towards reovirus act on early steps of replication that are already surpassed by time of ISG expression. It is also possible that infected and uninfected cells express distinct ISGs. In future, increased mechanistic insights into ISGs that impact reovirus, and single-cell sequencing to better understand ISG distribution between cells, will help clarify why T3D reovirus withstands antiviral signalling during the first round of replication.

## Reovirus σ3, a new determinant of RIG-I/IFN-independent gene expression

The S4-encoded σ3 of T3D$^{PL}$ was clearly associated with induction of RIG-I/ IFN-*in*dependent cellular genes. The σ3 protein has two well-characterized roles: it is the outer-most capsid

protein that maintains reovirus stability, and it binds dsRNA during virus replication to quench cellular dsRNA binding proteins such as RNA-dependent protein kinase (PKR) and subdues antiviral mechanisms [36, 38, 72–74]. Previously, we discovered that a tryptophan-to-arginine polymorphism at position 133 of σ3 is sufficient to promote replication of T3D$^{PL}$ relative to T3D$^{TD}$. When mapped on the 3D structure of σ3, the W133R polymorphism locates to the charged surface of σ3 involved in dsRNA binding [39, 74–79]. We therefore propose a model whereby T3D$^{PL}$ σ3 activates a dsRNA-dependent pathway such as PKR, distinct from MDA5 and RIG-I, to induce RIG-I/IFN-independent gene expression. The RIG-I/IFN-independent genes identified as group A by microarray analysis contain predicted binding sites for NF-κB transcription factors, and were classified as NF-κB dependent based on other transcriptomic analyses of LPS or coronavirus treated cells [80, 81]. Accordingly, activation of these genes by T3D$^{PL}$ likely involves NF-κB signalling, but in a manner that is distinct from RIG-I-induced NF-κB activation. For example, induction of RIG-I/IFN-independent genes might require distinct NF-κB subunits, or distinct partner transcription factors, or higher levels of NF-κB activation than RIG-I provides. Both PKR and NF-κB were previously found to be stimulated by reovirus and promote reovirus infection, for example by stimulating host protein shut-off, virus replication and apoptosis of infected cells [38, 46, 47, 82–84]. In future, a better understanding of how polymorphisms affect σ3 -dsRNA binding activities and stimulation of host factors such as NF-kB and PKR, will precisely resolve the role of σ3 in induction of RIG-I/IFN-independent genes.

## Materials and methods

### Cell lines

All cell lines were grown at 37˚C at 5% CO2 and all media was supplemented with 1x antibiotic antimycotics (A5955, Millipore Sigma). Except for NIH/3T3 media that was supplemented with 10% NCS (N4637, Millipore Sigma), all other media was supplemented with 10% FBS (F1051, Millipore Sigma). L929, A549, NIH/3T3, H1299, ID8, HCT 116 and B16-F10 cell lines (Dr. Patrick Lee, Dalhousie University), Huh7.5 (Dr. Michael Houghton, University of Alberta), WT MEFs, RIG-I/MDA5 -/- MEFs (Dr. Michael Gale Jr, University of Washington) and BHK-21-BSR T7/5 (Dr. Ursula Buchholz, NIAID) were generous gifts. L929 cell line was cultured in MEM (M4655, Millipore Sigma) supplemented with 1× non-essential amino acids (M7145, Millipore Sigma) and 1mM sodium pyruvate (S8636, Millipore Sigma). L929 cell line in suspension were cultured in Joklik's modified MEM (pH 7.2) (M0518, Millipore Sigma) supplemented with 2g/L sodium bicarbonate (BP328, Fisher Scientific), 1.2g/L HEPES (BP310, Fisher Scientific), 1× non-essential amino acids (M7145, Millipore Sigma) and 1mM sodium pyruvate (S8636, Millipore Sigma). A549, H1299, ID8 and T-47D cell lines were cultured in RPMI (R8758, Millipore Sigma). NIH/3T3, B16-F10, Huh7.5, WT MEFs, RIG-I/MDA5 -/-MEFs and BHK-21-BSR T7/5 cell lines were cultured in DMEM (D5796, Millipore Sigma) supplemented with 1mM sodium pyruvate (S8636, Millipore Sigma). BHK-21-BSR T7/5 cell line was selected in media containing 1mg/ml G418 (A1720, Millipore Sigma) every second passage. All cell lines were routinely assessed for mycoplasma contamination using 0.5µg/ml Hoechst 33352 (H1399, ThermoFisher Scientific) staining or PCR (G238, ABM).

### Reovirus stocks

Seed stock lysates of T3D-PL (Dr. Patrick Lee, Dalhousie University) and T3D-TD (Dr. Terence Dermody, University of Pittsburgh) were gifts in kind. Reovirus lysates were plaque purified and second or third passage L929 cell lysates were used as spinner culture inoculums.

## Immunocytochemistry (IHC)

In focus forming assays, 4% paraformaldehyde was added for 60 minutes (min) prior to removing overlays and fixing again with methanol. For cell monolayers, paraformaldehyde fixation was used alone. Cells were washed twice with PBS and incubated with blocking buffer (3% BSA/PBS/0.1% Triton X-100) for 1 hour at room temperature. Primary antibody (rabbit anti-reovirus pAb) diluted in blocking buffer at a final concentration of 1:10,000 was added and incubated overnight at 4˚C. Samples were washed 3 × 5minutes with PBS/0.1% Triton X-100. Secondary antibody diluted in blocking buffer (goat anti-rabbit AP for IHC) was added and incubated for 1-3hrs at room temperature. Samples were washed 3 × 5minutes with PBS/0.1% Triton X-100. For IHC, NBT (0.3mg/1ml) (B8503, Millipore Sigma) and BCIP (0.15mg/1ml) (N6639, Millipore Sigma) substrates diluted in AP buffer (100mM Tris pH 9.5, 100mM NaCl, 5mM MgCl2) were added and infected cells were monitored for black/purple staining using microscopy. Reactions were stopped with PBS/5mM EDTA. For FIA, nuclei were stained with 0.5μg/ml Hoechst 33352 (H1399, ThermoFisher Scientific) for 15min and stained samples were visualized and imaged using EVOS FL Auto Cell Imaging System (ThermoFisher Scientific).

## RNA extraction, RT-qPCR and HRM

Cells were lysed in TRI Reagent (T9424, Millipore Sigma) and the aqueous phase was separated following chloroform extraction as per TRI Reagent protocol. Ethanol was mixed with the aqueous phase and RNA isolation protocol was continued as per GenElute Mammalian Total RNA Miniprep kit (RTN350, Millipore Sigma) protocol. RNA was eluted using RNAse free water and total RNA was quantified using Biodrop DUO (Biodrop). Using 1μg RNA per 20μl reaction, cDNA synthesis was performed with random primers (48190011, ThermoFisher Scientific) and M-MLV reverse transcriptase (28025013, ThermoFisher Scientific) as per M-MLV reverse transcriptase protocol. Following a 1/8 cDNA dilution, RT-qPCR reactions were executed as per SsoFast EvaGreen Supermix (1725204, Bio-Rad) protocol using a CFX96 system (Bio-Rad). All RT-qPCR reaction plates included negative controls; no template and no reverse transcription controls.

Oligonucleotides used for RT-qPCR reactions are as follows: Reovirus S4 [Forward GGAA CATTGTGAGAGCAGCA, reverse GCAAGCTAGTGGAGGCAGTC], Reovirus M2 [Forward ACGATGTCCCCACTATCAGC, reverse GATTGCTTCGGCTATCTTCG], mGAPDH [Forward TGGCAAAGTGGAGATTGTTGCC, reverse AAGATGGTGATGGGCTTCCCG], mIfnb1 [Forward CCCTATGGAGATGACGGAGA, reverse CTGTCTGCTGGTGGAGTTC A], mIfna4 [Forward CTGCTGGCTGTGAGGACATA, reverse AGGAAGAGAGGGCTCTC CAG], mRigI [Forward GCCCCTACTGGTTGTGGAAA, reverse GTGAGAACACAGTTG CCTGC], mRsad [Forward TTGACCACGGCCAATCAGAG, reverse TTGACCACGGCCAA TCAGAG], mMx1 [Forward TCTGTGCAGGCACTATGAGG, reverse ACTCTGGTCCCCA ATGACAG], mCxcl10 [Forward ATGACGGGCCAGTGAGAATG, reverse CATCGTGGCA ATGATCTCAACA], mCcl4 [Forward CAGCTGTGGTATTCCTGACCAA, reverse AGCTG CTCAGTTCAACTCCA], mCcl5 [Forward ATATGGCTCGGACACCACTC, reverse TCCTT CGAGTGACAAACACG], mCxcl1 [Forward ACCGAAGTCATAGCCACACTC, reverse CTCCGTTACTTGGGGACACC], mCxcl2 [Forward CCCAGACAGAAGTCATAGCCAC, reverse TGGTTCTTCCGTTGAGGGAC], mFas [Forward GTCCTGCCTCTGGTGCTTG, reverse AGCAAAATGGGCCTCCTTGA], mCsf2 [Forward ATGCCTGTCACGTTGAATGA, reverse CCGTAGACCCTGCTCGAATA]

For HRM analysis, RNA extraction, cDNA synthesis and qPCR were conducted as above but using S4 primers that flanks polymorphisms between T3D strains [Forward: GCGTTCAA TGGTGTGAAAC, Reverse GGCCCTTCGATGGATCA]. Melt curve analysis was performed

from 60˚C to 95˚C at increments of 0.1˚C for 5 seconds per temperature followed by fluorescence measurement.

## Western blot analysis

Cells were washed with PBS and lysed in RIPA buffer (50mM Tris pH 7.4, 150mM NaCl, 1% IGEPAL CA-630 (NP-40), 0.5% sodium deoxycholate) supplemented with protease inhibitor cocktail (11873580001, Roche) and phosphatase inhibitors (1mM sodium orthovanadate, 10mM β-glycerophosphate, 50mM sodium fluoride). For each 12 well, 100μl lysis buffer was used, and volume was scaled up/down according to well size. Following addition of 5× protein sample buffer (250mM Tris pH 6.8, 5% SDS, 45% glycerol, 9% β-mercaptoethanol, 0.01% bromophenol blue) for a final 1× protein sample buffer, samples were heated for 5min at 100˚C and loaded onto SDS-acrylamide gels. After SDS-PAGE, separated proteins were transferred onto nitrocellulose membranes using the Trans-Blot Turbo Transfer System (Bio-Rad). Membranes were blocked with 3% BSA/TBS-T (blocking buffer) and incubated with primary and secondary antibodies (5ml blocking buffer, primary and secondary antibody per mini (7×8.5cm) membrane). Membranes were washed 3×5min with TBS-T after primary and secondary antibodies. (10ml wash buffer per mini (7×8.5cm) membrane). Membranes with HRP-conjugated antibodies were exposed to ECL Plus Western Blotting Substrate (32132, Thermo-Fisher Scientific) (2ml substrate per mini (7×8.5cm) membrane) for 2min at room temperature. Prior to visualization, membranes were rinsed with TBT. Membranes were visualized using ImageQuant LAS4010 imager (GE Healthcare Life Sciences), and densitometric analysis was performed by using ImageQuant TL software (GE Healthcare Life Sciences). Note that a "wash" in the above protocol constitutes addition of reagent, gently rocking back and forth twice and removal of reagent.

## Reovirus plaque assays

Reovirus dilutions were added to 100% confluent L929 cells for 1hr at 37˚C with gentle rocking every 10 minutes, followed by addition of agar overlay (1:1 ratio of 2% agar 2× JMEM media). Overlays were allowed to solidify for 30 minutes at room temperature and incubated at 37˚C. When plaques became visible (3–7 days post infection), 4% formaldehyde solution (Alfa Aesar) was added to the overlay for 30 minutes. Formaldehyde was discarded, agar overlays were carefully scooped out and cells were further fixed with methanol for 5min. Methanol was discarded, and cells were stained with crystal violet solution (1% crystal violet (Fisher Scientific) in 50% ethanol and 50% water) for 10min and rinsed with water. For cell lines other than L929, after methanol staining, plaques were stained using immunocytochemistry with rabbit anti-reovirus pAb. Plaques were scanned on the ImageQuant LAS4010 imager (GE Healthcare Life Sciences), and plaque area was measured using ImageQuant TL software (GE Healthcare Life Sciences).

## Flow cytometry

Cell culture media was aspirated, and cell monolayers were rinsed with PBS. PBS was discarded and trypsin (200μl/12well) was added and incubated until cells detached. Tryspin in detached cells was quenched with cell culture media (1ml/12well). Volumes of PBS, trypsin and cell culture media were scaled up/down according to well size. Detached cells were centrifuged and washed with PBS. PBS was aspirated, cell pellet was gently resuspended in 4% paraformaldehyde and incubated at 4˚C for 30min. Samples were centrifuged and washed with 1ml PBS/0.1% Triton X-100 (wash buffer). Wash buffer was aspirated, cell pellet was gently resuspended in 3% BSA/wash buffer (blocking buffer) and incubated at 1 hour at room

temperature for 30min. Samples were spiked with primary antibody (rabbit anti-reovirus pAb) diluted in blocking buffer for a final concentration of 1:10,000 and incubated overnight at 4°C. Samples were centrifuged and washed twice with 1ml wash buffer. Wash buffer was aspirated, and cell pellet was gently resuspended in secondary antibody diluted in blocking buffer (goat anti-rabbit Alexa Fluor 647 at 1:1:2,000 dilution) and incubated for 1-3hrs at room temperature. Samples were centrifuged and washed twice with 1ml wash buffer. Wash buffer was aspirated, and cell pellet was gently resuspended in 500μl PBS. Samples were processed using a FACSCanto (BD Biosciences) and data was analyzed using FSC Express 5 (De Novo Software). Total cells were gated using FSC-A and SSC-A, while single cells were gated using FSC-A and FSC-H. A minimum of 10,000 total cells were collected for each sample. Note that a "wash" in the above protocol constitutes aspiration of supernatant, resuspension of pellet and centrifugation. Prior to fixation, cells were centrifuged at 500g for 5min at 4°C. After fixation, cells were centrifuged at 1,000g for 5min at 4°C.

## Lentivirus production and generation of stable cell lines expressing shRIG-I, shEMPTY or shSCR

Lentivirus was generated as per MISSION Lentiviral Packaging Mix protocol (Millipore Sigma) using 6 well plates. Lentivirus containing supernatants were collected every 12hrs for 72hrs. Pooled lentivirus collections were centrifuged at 600g for 10min at 4°C, and the supernatant was 0.45μm filter sterilized, aliquoted, and stored at -80°C.

Lentivirus stock was diluted in cell culture media supplemented with sequabrene (Millipore Sigma) at 8μg/ml final. Dilutions ranged from 1/3 to 1/36. Media was aspirated from 12 well plates with cells at 50–60% cell confluency, 500μl lentivirus dilution was added to each well and allowed to incubate at 37°C for 12hrs. Lentivirus was aspirated and replaced with cell culture media for an additional 12-24hrs until cells became confluent. At 100% confluency, cells were trypsinized, transferred to a 6 well plate and incubated at 37°C for 12hrs. Cell culture media was replaced with media supplemented with puromycin (2μg/ml for NIH/3T3 cell line) and cell death was monitored by microscopy. Cells not treated with lentivirus but exposed to puromycin supplemented media were used to determine when untransduced cells were killed. When lentivirus transduced cells reached 90–100% confluency, cells, were trypsinized and transferred to 55cm2 flask with media supplemented with puromycin. Lowest lentivirus dilution with minimal (0–10%) cell death was selected for further assessment. Puromycin selection was performed every second passage.

## Microarray analysis

NIH/3T3 cells were transduced with lentivirus generated from either plko.1 vector containing shRNA targeting mouse RIG-I (shRIG-I), mouse IFN receptor (shIFNAR), or with retrovirus pBabe vector containing mouse RIG-I full length ORF (mRIG-I O/E). Stably transduced cells were selected with 2μg/ml puromycin for seven days, followed by T3D$^{PL}$ infection at the dose empirically determined to cause 70% reovirus-antigen-positive cells on NIH/3T3 shGFP cells by immunofluorescence with anti-reovirus polyclonal antibodies. At 12 hpi, RNA was extracted from reovirus infected and mock infected samples and sent for microarray processing at MOgene LC (1005 North Warson Road, Suite 403, St. Louis, MO 63132 USA) using the Agilent Mouse SurePrint G3 GE, 8x60k – 028005 array type. Raw microarray image files were extracted with Agilent's Gene Expression Microarray Software, GeneSpring GX and is accessible from NCBI GEO DataSet # GSE127850.

For each data set, the normalized microarray data along with gene and sample information was input into the pheatmap package in R. Density graphs were plotted using ggplot package

and the Venn diagrams were made using powerpoint. Gene lists were compiled in excel and are available as supplemental figures.

## Supporting information

**S1 Fig. Key characteristic IFN differences between T3D$^{PL}$ and T3D$^{TD}$ are reproduced in human and murine cancer cell lines. (A-E)** Cells were infected with T3D$^{PL}$ or T3D$^{TD}$ and samples were collected at 12hpi for immunocytochemical staining of reovirus infected cells (A), Western blot analysis of reovirus proteins (μ1, σ3), IRF3, P-IRF3, β actin (B) or RNA extraction followed by cDNA synthesis and RT-qPCR for the indicated genes (C-E). **(A-C)** A549 cells. **(D)** B16-F10 cells. **(E)** ID8 cells. All genes except Reovirus S4/M2 were normalized to MOCK. Reovirus S4/M2 values were normalized to the lowest T3D$^{TD}$ dose. Each point represents a biological replicate n = 1–2. **(F)** Similar to Fig 5A, RIG/IFN-independent Csf2 mRNA levels relative to housekeeping gene GAPDH was quantified using RT-qPCR, in WT or RIG-I/MDA5 -/- double knockout (DKO) MEFs infected with T3D$^{PL}$ or T3D$^{TD}$ at MOI 6 for 12hpi. Values were normalized to MOCK WT MEF. Each point represents a technical replicate for n = 2 independent experiments.
(TIF)

**S2 Fig. IFN signalling has minimal impact on initial reovirus infection. (A)** L929 cells were treated with 1000 U/ml/12well of purified IFNβ for 18hrs at 37˚C. Samples were collected for RNA extraction, cDNA synthesis and RT-PCR using gene-specific primers (corrected for GAPDH). Values were standardized to untreated sample. **(B-C)** L929 cells were treated with IFNβ at the indicated timepoints and/or infected with T3D$^{PL}$ or T3D$^{TD}$ for 18hrs. Samples were collected and processed for viral titres (B) and Western blot analysis (C). Protein samples in (C) were quantified using densitometric band analysis with an ImageQuantTL. Each point represents a biological replicate.
(TIF)

**S1 Table. Whole genome microarray data.** The excel spreadsheet summarizes all genes expression data described in Fig 6, includeing genes grouped in categories A-H as separate sheets. The data was also submitted to the reposatory indicated in the material and methods, but the excel sheet should hopefully assist readers in rapidly checking "their favorite gene" in the dataset.
(XLSX)

## Acknowledgments

We would like to thank Kevin Coombs at the University of Manitoba and Terrence Dermody at University of Pittsburgh for generously sharing their laboratory reovirus T3D virus lysates, Aja Reiger at the Faculty of Medicine and Dentistry flow cytometry facility, Rob Maranchuk at the Li Ka Shing Institute of Virology RNAi screening facility and Stephen Ogg at the Faculty of Medicine & Dentistry cell imaging centre, for valuable technical advice, training and support. We appreciate all the helpful discussions and suggestions by members of the Maya Shmulevitz, David Evans, Mary Hitt, Ronald Moore and Patrick Lee laboratories.

## Author Contributions

**Conceptualization:** Adil Mohamed, Maya Shmulevitz.

**Data curation:** Adil Mohamed, Prathyusha Konda, Maya Shmulevitz.

**Formal analysis:** Adil Mohamed, Prathyusha Konda, Maya Shmulevitz.

**Funding acquisition:** Maya Shmulevitz.

**Investigation:** Adil Mohamed, Maya Shmulevitz.

**Methodology:** Adil Mohamed, Maya Shmulevitz.

**Project administration:** Shashi Gujar, Maya Shmulevitz.

**Resources:** James R. Smiley.

**Software:** Prathyusha Konda.

**Supervision:** Maya Shmulevitz.

**Validation:** Adil Mohamed, Heather E. Eaton.

**Visualization:** Adil Mohamed, Prathyusha Konda, Maya Shmulevitz.

**Writing – original draft:** Adil Mohamed, Maya Shmulevitz.

**Writing – review & editing:** Adil Mohamed, Prathyusha Konda, Shashi Gujar, James R. Smiley, Maya Shmulevitz.

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
