## [Decision Letter · Decision Letter 0]

21 Feb 2020

Dear Dr. Shmulevitz,

Thank you very much for submitting your manuscript "Closely related reovirus lab strains induce opposite expression of RIG-I/IFN-dependent versus -independent host genes, via mechanisms of slow replication versus polymorphisms in dsRNA binding σ3 respectively" (PPATHOGENS-D-20-00031) for consideration at PLOS Pathogens. As with all papers peer reviewed by the journal, your manuscript was reviewed by members of the editorial board and by several independent peer reviewers. Based on the reports, we regret to inform you that we will not be pursuing this manuscript for publication at PLOS Pathogens.

Although the reviewers generally expressed enthusiasm for the study, several major issues were noted, some of which will require additional experiments. The following concerns with the manuscript need to be addressed.

1. It is possible that variation in the particle:PFU ratios between virus preps could influence IFN induction. This was of particular importance to the testing of the reassortant viruses.

2. There was no direct test of whether IFN priming underlies differences between the initial and subsequent rounds of replication by pretreating with IFN.

3. It could not be discerned whether the cells in co-infection experiments were actually infected by by both viruses. 

4. Related to the co-infection experiments, how does establishment of separate replication centers by fast- and slow-replicating viruses allow for reassortment?

5. The relationship to the oncolytic properties of reovirus is a lynchpin of the rationale for the study and key to the broader impact of the work. However, the vast majority of the results were obtained in L929 cells and the relevance of the results to oncolysis is unclear.

6. The presentation of some data may be difficult for readers to interpret (see specific comments from Reviewer 3).

7. Statistical analysis is missing from certain figures.

The reviewers also made a number of suggestions that would be helpful in clarifying the key points of the study.

The reviews are attached below this email, and we hope you will find them helpful if you decide to revise the manuscript for submission elsewhere. We are sorry that we cannot be more positive on this occasion. We very much appreciate your wish to present your work in one of PLOS's Open Access publications. 

Thank you for your support, and we hope that you will consider PLOS Pathogens for other submissions in the future.

Sincerely,

Jing-hsiung James Ou

Section Editor

PLOS Pathogens

Kasturi Haldar

Editor-in-Chief

PLOS Pathogens

orcid.org/0000-0001-5065-158X

Michael Malim

Editor-in-Chief

PLOS Pathogens

orcid.org/0000-0002-7699-2064

Reviewer's Responses to Questions

**Part I - Summary**

Reviewer #1: Overall, I think the findings presented by Adil Mohamed et al. are of sufficient novelty and interest for the RNA virology community at large. However, I do think that the paper would benefit from further clarifications and possible additional experiments. While I knowledge and value the hard-work from the authors, here are my comments, which are intended to further strengthen the manuscript.

Reviewer #2: The authors investigate the differential effects of two laboratory variants of the reovirus prototype strain type 3 Dearing (T3D) on host cell innate immune signaling. They demonstrate that despite slower replication kinetics (at both the RNA and protein level), the T3D-TD variant induces higher levels of IFN and ISG mRNA than the T3D-PL variant, based on polymorphisms in the S4, M1, and L3 gene segments. Experiments that decrease the replication kinetics of T3D-PL using inhibitors of RNA and/or protein synthesis result in enhanced IRF-3 phosphorylation, suggesting that the rate of replication the exposure of viral PAMPs to cellular pattern recognition receptors. Interestingly, in co-infection experiments, the T3D-TD strain was phenodominant, in that high levels of IRF-3 phosphorylation were observed, despite robust replication of T3D-PL in comparison to T3D-TD. The authors utilized either RIG-I/Mda-5 double knockout (DKO) fibroblasts or shRNA-mediated knockdown of RIG-I to demonstrate that these pathways did not impact reovirus infectivity or early rounds of viral RNA or protein synthesis, despite their efficacy in reducing induction of IFNs and ISGs. The authors then conducted a whole genome microarray analysis in NIH3T3 cells over- or under-expressing RIG-I or under-expressing the IFN receptor. These experiments, when repeated in other cell lines, identified a number of IFNR-independent, RIG-I-independent genes that were upregulated by reovirus infection, and interestingly, this subset of genes were also differentially induced by the T3D strains, with T3D-PL inducing higher levels than T3D-TD (in contrast to the RIG-I/IFN-dependent genes). Finally, the authors confirm that polymorphisms in the S4 gene were responsible for the differences in expression of a subset of these RIG-I/IFN-independent genes. 

Overall, the manuscript presents intriguing data highlighting the sharp differences in cellular responses that can be elicited by very closely related virus strains, which has significant implications for the use of these strains as potential oncolytic therapies. The data is generally clear and well-controlled and the manuscript is generally well written.

Reviewer #3: This study examined the differences in interferon (IFN) induction between two closely related isolates of T3D reovirus (T3D-PL and T3D-TD). Here it is shown that the PL isolate replicates quickly and lower levels of IFN and IFN-stimulated genes (ISGs) are detected in cells, whereas the TD isolate replicates more slowly and higher levels of IFN and IFN-stimulated genes are detected. Co-infection experiments demonstrated that the lower levels of IFN production in cells infected with the PL isolate did not appear to be due to a virally encoded IFN antagonist. The slower replicating T3D-TD appeared to induce RIG-I-dependent ISG expression, whereas the rapidly replicating T3D-PL appeared to induce RIG-I-independent genes. The role of any host gene on enhancing or inhibiting T3D reovirus replication was not examined in detail; thus, a large data set is presented but the impact or inhibitory mechanisms were not further explored. 

The rationale at the beginning of this study seems to be to examine the reason for faster replication kinetics of T3D-PL, and to tie that to its oncolytic properties, but that idea never comes to fruition. In fact, previous studies already show that the replication kinetics are cell-independent. Therefore, it is unclear if any of the cell signaling responses described in this manuscript directly influence the T3D viral life cycle or its oncolytic properties.

**Part II – Major Issues: Key Experiments Required for Acceptance**

Reviewer #1: 1. Although the authors showed some data in other mouse cell lines (B16-F10 and NIH3T3) in Fig 6, all the data presented is representative of a single mouse cell line, L929. The authors claimed that the study presented herein could be informative for the oncolytic reovirus community and the implementation of better oncolytic reoviruses for cancer treatment in a future. However, most of the data is generated in L929 cells, which are normal mouse fibroblasts, there is not data in human cancer cell lines. These are issues that would affect the translational impact of the findings, I do believe that lack of data related to this does limit somewhat the impact of the study”

2. Fig 1. The figure panels are not properly indicated in the text: Line 154 (should say Fig 1D), line 158 (should say Fig 1E) and line 159 correspond to Fig 1F. 

3. Fig 1 legend (line 1090). Was the western blot performed with a �2 or a �1 antibody? There is a discrepancy between text and figure. 

4. Fig 2B-C. Are there other factors in the genome of T3DPL important for the observed phenotype. How do the double and triple-reassortants perform in these qPCR assays?

5. Fig 2D and line 197. It is hard to conclude that the mono-ressortant strains have an intermediate phenotype (especially if the plaque assay the dilution factor used was 1:10). The variation observed between the ressortants and the parental T3DTD could be just dues to titer error. Quantification of viral genomes by qPCR could be more informative. Also, stats and error bars should be indicates. 

6. Fig 2 legend. P values are not indicated. 

7. How does T3DPL “performs” in cells that have been pre-treated with type I interferons?. In vivo, for example in a tumor, it is expected that the main source of interferons indeed are not the infected cells but the immune cells that react to the infection. 

8. Please clarify the point made in lines 335-337. I thought that T3DPL was a better oncolytic, but herein you imply that T3DTD is better because induces interferon which can lead to tumor clearance?

9. Fig 5. The data shown in panel C seems in disagreement with the data shown in panel H. MEF-DKO show lower T3DPL S4 levels compared to WT MEFs, but in panel H, RIG-I KD cells show higher T3DPL S4 levels compare to control. 

10. In the experiment where co-infections between T3DPL and T3DTD viruses were performed, how we can ensure that indeed “co-infections” happened and that the signalling events triggered independently by these two viruses indeed happen in “parallel’. I think the robustness of the conclusions and model proposed could strongly benefit by performing single cell sequencing. 

11. Why T3DPL does not activate IRF phosphorylation and IFNs?

12. Indicate in all figure legends number of biological replicates and error bars and statistics have to be included when appropriate.

Reviewer #2: One overarching concern not addressed in the methods section, or in the experiments themselves, is the variability of different preparations of virus stocks. There is precedent for considerable variability in the immunoreactivity of different reovirus stock preparations, particularly with regard to variations in the particle:PFU ratio. The authors tangentially mention the effects of defective virions in activating innate immune signaling in both the introduction and discussion, but do not examine these concerns themselves. If, for example, the T3D-TD preparations have higher levels of defective particles that could deliver PAMPs to PRRs, that could account for greater innate immune signaling despite overall lower levels of viral RNA synthesis in the infected cell. A robust examination of the particle:PFU ratio of the stocks used would allay these concerns; similarly, if the experiments were repeated across multiple viral preparations, this would strongly suggest that it is a phenotype related to the cell biology of the infection, rather than the result of the initial inoculum. Evidence from Stuart et al also suggests that differential delivery of genomic dsRNA (possibly via differences in capsid stability during uncoating) underpins differences in IRF-3 activation, which is not addressed by the authors. A second critical question is the location of virus factories in co-infected cells; the authors postulate a model whereby each entering core sets up its own virus factory, thereby keeping the different strains “segregated” in the cytoplasm, which facilitates i) the rapid replication of T3D-PL in its factories and ii) the detection of T3D-TD PAMPs in its factories. However, this model would tend to preclude the possibility of the production of reassortant viruses. While directly testing this (perhaps using FISH probes specific to distinct viral polymorphisms?) might be outside the scope of the current manuscript, directly demonstrating that the two virus strains elicit distinct factory compartments would certainly increase the overall significance of the work. A final major suggestion would be to differentiate between plus- and minus-strand viral RNA in measuring levels of RNA synthesis in the experiments; the authors use randomers to facilitate cDNA synthesis, which presumably would equivalently amplify both viral mRNA as well as negative-sense genomic RNA. Using a reovirus specific primer (either forward or reverse to amplify negative-sense or positive-sense RNA respectively) would allow for the more fine-grained analysis of whether the differences between strains in RNA production is solely in producing mRNAs vs. producing genomic mRNA during the production of viral progeny particles.

Reviewer #3: 1a. When evaluating IFN or ISGs by RT-qPCR, it is most common to set levels relative to the mock sample. By doing so, it is easier for the reader to evaluate and understand if levels have increased during viral infection. In this manuscript, the authors have chosen to set all samples relative to T3D-TD infection at MOI 1 – please provide an explanation for doing so. 

1b. In setting T3D-TD at MOI 1 as the reference, it becomes difficult for the reader to determine how much higher levels of IFN or ISGs were in comparison to mock. In this situation, the authors must be exceedingly careful as to the conclusions being drawn. For instance, it is reasonable to state that in T3D-TD infected cells, the levels of Ifnb1 at MOI 9 are about 7-fold higher than at MOI 1, but from the data presented it cannot be determined if there is a measurable reduction in Ifnb1 levels comparing T3D-TD at MOI 1 to T3D-PL at MOI 1 (Fig. 1F). The authors then conclude that T3D-TD induced higher expression of ISGs Mx1 and Rsad2 relative to T3D-PL (line 160), yet Fig. 1G appears to show a less than 2-fold change in Rsad2 and Mx1 levels. The suggestion that T3D-TD triggered greater IFN-response signaling pathways (lines 161-162) does not appear to be supported by the data presented in Fig. 1G.

2a. Although there is a complex attempt to show that IFN did not affect the first round of T3D-TD replication, the most direct experiment would use IFN treatment (pre-infection and/or post-infection) to show that IFN does not impact the initial round of T3D-TD replication. 

2b. The authors suggest that the slower rate of T3D-TD replication causes greater IFN and ISG production. Therefore, there is an examination of mRNA levels of numerous ISGs, but there is no direct demonstration that any of them impact T3D-TD titers or spread. Which of the examined ISGs are relevant to the reovirus system? Do any of them inhibit T3D-TD or T3D-PL spread from cell to cell? 

3. Several figures are lacking statistical analysis, which will be important to supporting some of the claims in the manuscript. Figs. 1E, F, G absolutely require statistical analysis. Fig. 2D is lacking error bars and statistical analysis. Fig. 5 and Fig. 7 are also lacking statistical analysis. 

4. The use of the word “replication” in this manuscript is rather vague. When discussing reovirus infections, typically replication refers to the specific step of dsRNA synthesis. However, in this manuscript replication appears to refer to the entire life cycle from entry to nascent virion production. It would offer some clarity for more specific terminology to be used.

**Part III – Minor Issues: Editorial and Data Presentation Modifications**

Reviewer #1: Minor comments

Line 64- did the authors meant “it is not surprising”?

Line 139- impact word used twice

Line 205- should say “production”

Line 240-typo (phosphor-IRF3)

Line 331- the authors meant: “T3D PL (not T3DTD) inherently replicates better than T3DTD?

Line 383- Typo “Were strongly..”

Reviewer #2: 1) Line 76 should read “burst size” not “bust size”

2) Line 87 should read “distinct” not “district”

3) Line 158-160: References to specific figure 1 panels in the text do not match with the specific panels in the figure

4) Line 326: Do the authors mean that “IFN signaling does NOT account…”?

Reviewer #3: 1. There is strong support for replication of data from previous studies, but Fig. 1C and 2A appear to be replicated data previously published by the Shmulevitz lab. In a data-dense paper such as this one, it is okay to just refer to the previous studies to reduce the amount of data presented. Additionally, drawings representing well-known signaling pathways are not necessary to include. 

2. Please note in the Materials and Methods if no template and no reverse transcriptase controls were included in every RT-qPCR setup. 

3. There are several instances of references to figures in the text that do not match: for instances lines 158-159 refer to Fig. 1D, but the data is found in Fig. 1F; another instance in line 160 refers to Fig. 1E but the data is found in Fig. 1G. Please carefully make sure all figures are correctly referenced in the text. 

4. Line 331 reads, “…T3D-TD inherently replicates better than T3D-TD…” – please correct.

PLOS authors have the option to publish the peer review history of their article (what does this mean?). If published, this will include your full peer review and any attached files.

Reviewer #1: No

Reviewer #2: No

Reviewer #3: No

---

## [Decision Letter · Decision Letter 1]

13 Jul 2020

Dear Dr. Shmulevitz,

We are pleased to inform you that your manuscript 'Closely related reovirus lab strains induce opposite expression of RIG-I/IFN-dependent versus -independent host genes, via mechanisms of slow replication versus polymorphisms in dsRNA binding σ3 respectively' has been provisionally accepted for publication in PLOS Pathogens.

Best regards,

Karl Boehme, PhD

Guest Editor

PLOS Pathogens

Jing-hsiung James Ou

Section Editor

PLOS Pathogens

Kasturi Haldar

Editor-in-Chief

PLOS Pathogens

orcid.org/0000-0001-5065-158X

Michael Malim

Editor-in-Chief

PLOS Pathogens

orcid.org/0000-0002-7699-2064

Reviewer Comments (if any, and for reference):

Reviewer's Responses to Questions

**Part I - Summary**

Reviewer #2: The authors have adequately addressed reviewer concerns, and therefore I recommend the manuscript for publication.

Reviewer #3: This is a revised manuscript exploring differences in replication kinetics and interferon (IFN) induction between two closely related isolates of T3D reovirus (T3D-PL and T3D-TD). While it is not unexpected that IFN induction inversely correlates with reovirus replication, this study carefully narrows the cause of the observed differences to a single amino acid in the sigma3 protein. The authors have invested considerable effort to address the previous comments of three reviewers. The additional data has strengthened the manuscript; particularly noteworthy is the addition of pre- and post-infection treatments with IFN, and the use of several different cell lines for certain experiments.

**Part II – Major Issues: Key Experiments Required for Acceptance**

Reviewer #2: (No Response)

Reviewer #3: None

**Part III – Minor Issues: Editorial and Data Presentation Modifications**

Reviewer #2: (No Response)

Reviewer #3: None

PLOS authors have the option to publish the peer review history of their article (what does this mean?). If published, this will include your full peer review and any attached files.

Reviewer #2: No

Reviewer #3: No

---

## [Editor Report · Acceptance letter]

15 Sep 2020

Dear Dr. Shmulevitz,

We are delighted to inform you that your manuscript, "Closely related reovirus lab strains induce opposite expression of RIG-I/IFN-dependent versus -independent host genes, via mechanisms of slow replication versus polymorphisms in dsRNA binding σ3 respectively," has been formally accepted for publication in PLOS Pathogens.

Best regards,

Kasturi Haldar

Editor-in-Chief

PLOS Pathogens

orcid.org/0000-0001-5065-158X

Michael Malim

Editor-in-Chief

PLOS Pathogens

orcid.org/0000-0002-7699-2064